# Point Cloud Self-supervised Learning via 3D to Multi-view Masked Learner

## Abstract

Recently, multi-modal masked autoencoders (MAE) has been introduced in 3D self-supervised learning, offering enhanced feature learning by leveraging both 2D and 3D data to capture richer cross-modal representations. However, these approaches have two limitations: (1) they inefficiently require both 2D and 3D modalities as inputs, even though the inherent multi-view properties of 3D point clouds already contain 2D modality. (2) input 2D modality causes the reconstruction learning to unnecessarily rely on visible 2D information, hindering 3D geometric representation learning. To address these challenges, we propose a 3D to Multi-View Learner (Multi-View ML) that only utilizes 3D modalities as inputs and effectively capture rich spatial information in 3D point clouds. Specifically, we first project 3D point clouds to multi-view 2D images at the feature level based on 3D-based pose. Then, we introduce two components: (1) a 3D to multi-view autoencoder that reconstructs point clouds and multi-view images from 3D and projected 2D features; (2) a multi-scale multi-head (MSMH) attention mechanism that facilitates local-global information interactions in each decoder transformer block through attention heads at various scales. Additionally, a novel two-stage self-training strategy is proposed to align 2D and 3D representations. Empirically, our method significantly outperforms state-of-the-art counterparts across various downstream tasks, including 3D classification, part segmentation, and object detection. Such performance superiority showcases that Multi-View ML enriches the model's comprehension of geometric structures and inherent multi-modal properties of point clouds.

## 1 Introduction

3D vision is a critical research area with applications in autonomous driving (Duan et al., 2019; Liu et al., 2023; Li et al., 2022) and robotics (Zou et al., 2021; Zhou et al., 2021), thanks to its ability to model and interpret the real world. However, manually collecting and annotating 3D point cloud data is both expensive and time-consuming, posing significant challenges for developing effective 3D models. Consequently, self-supervised learning (SSL) of 3D representations has emerged as a key research direction.

Recent works have leveraged 2D images to enhance 3D self-supervised learning (Guo et al., 2023; Chen et al., 2023; Qi et al., 2023; Wang et al., 2023). Specifically, methods such as Joint-MAE (Guo et al., 2023) and PiMAE (Chen et al., 2023) utilize masked images and point clouds as inputs to reconstruct complete images and point clouds. However, we argue that using both 2D and 3D modalities as inputs in these MAE-based methods presents two significant drawbacks. **First**, incorporating both 2D and 3D modalities as input for training is redundant and inefficient. Point clouds inherently encapsulate multi-modal data, as they can be directly translated into multi-view depth images or rendered as 2D images. This eliminates the necessity for input images for 2D reconstruction. **Secondly**, using 2D images as input may cause the network to rely heavily on visible 2D information when predicting masked regions. Specifically, when 2D view information is directly provided, it inadvertently "leaks" viewpoint semantic information to the network. Consequently, the underlying multi-view geometric information is neglected, preventing the network from developing a comprehensive understanding of how the 2D view should appear from different angles. As a result, the model is unable to develop multi-view geometric representations, which is essential for effective

3D representation learning, as highlighted in previous studies (Hamdi et al., 2021b;a; Su et al., 2015; Robert et al., 2022).

To address these limitations, we propose leveraging the intrinsic multi-modal nature of point clouds without the redundant use of input images, thereby enhancing the efficiency and effectiveness of 3D self-supervised learning. More formally, we introduce an innovative method called Multiview Masked Learner (Multiview-ML), designed to extract the multi-view 2D information inherently embedded within 3D point clouds, thus advancing 3D representation learning. Specifically, we input point clouds into an initial encoder to obtain intermediate 3D encoded tokens. Using the provided pose information, we then project these intermediate 3D tokens onto 2D image tokens through our proposed view-based feature projection (see Figure 2a). The projected multi-view 2D tokens and the original 3D tokens, augmented with modality, positional, and pose embeddings, are then fed into a fusion encoder to obtain the corresponding encoded 3D and multi-view 2D features. Subsequently, we pass these features into two separate decoders to independently reconstruct the complete multi-view 2D images and 3D point clouds.

Furthermore, to enhance the integration of local attention mechanisms for reconstruction, we introduce a Multi-Scale Multi-Head (MSMH) attention mechanism—a simple yet effective module that provides broader local and global attention. Unlike previous standard attention mechanisms, MSMH organizes tokens into distinct, non-overlapping local groups. Self-attention is then applied within each subgroup rather than across all individual tokens. Additionally, we implement a multi-scale design with varying group sizes, allowing smaller groups to capture fine-grained local details while larger groups capture broader global context. Finally, we concatenate the multi-scale attention features to ensure the model effectively acquires both local and global information.

Moreover, unlike previous methods (Pang et al., 2022; Guo et al., 2023), which primarily focus on reconstructing raw masked inputs, our approach involves reconstructing masked 2D and 3D representations within a latent space, drawing inspiration from the SimSiam (Chen & He, 2021). We propose a two-stage training procedure to enhance multi-modal masked representation learning: *Stage One*: 3D to multi-view autoencoder (teacher model) takes the complete point cloud as input and outputs both the reconstructed point cloud and multi-view images. The autoencoder extracts latent features that contain rich geometric information, effectively restoring the point cloud and multi-view images. *Stage Two*: We introduce a student network based on a masked autoencoder, which leverages masked point clouds to predict features generated by the teacher encoder in the first stage. Notably, our objective function is designed to predict latent space features instead of 3D point clouds. This ensures that the student model learns well-aligned and contextualized representations across modalities.

We conducted extensive experiments to validate our Multiview-ML approach. Our method significantly outperforms previous approaches across downstream tasks such as 3D shape classification, part segmentation, and object detection, underscoring its effectiveness in learning robust 3D geometric representations. A key insight from our study is *the limited effectiveness of using 2D images as input for 3D geometric learning through MAE*. We believe this finding presents an intriguing opportunity in the design space for developing learning strategies for 3D representation learning. Our key contributions are summarized as follows:

⋄ We propose a 3D to multi-view autoencoder that reconstructs point clouds and multi-view images solely from 3D point clouds, eliminating the need for additional input images and enhancing the learning of 3D geometric features.

⋄ We propose a Multi-Scale Multi-Head (MSMH) attention mechanism that integrates local and global contextual information by organizing distinct, non-overlapping local groups at multiple scales within the reconstructed features.

⋄ We develop a two-stage training strategy for multi-modality masked feature prediction, enhancing the alignment between 2D and 3D representations by having the student network predict the teacher's latent features from masked point clouds.

⋄ Our method outperforms existing approaches across various downstream tasks, underscoring the importance of leveraging the inherent multi-view 2D information present in point clouds for effective 3D representation learning.

## 2 RELATIVE WORK

**Masked Autoencoder for Point Cloud**. MAE (He et al., 2022) masks random patches of an input image and reconstructs the missing pixels, effectively learning modality-specific representations. This approach has achieved significant success across various domains (Devlin et al., 2018; Radford et al., 2019a;b). Recently, Point-BERT (Yu et al., 2022) and Point-MAE (Pang et al., 2022) adapted MAE to reconstruct masked 3D point clouds. Furthermore, Zhang et al.(Zhang et al., 2022) proposed PointM2AE, which uses pyramid architectures to capture both fine-grained and high-level semantic features of 3D shapes. However, these methods primarily focus on a single 3D modality. In contrast, multi-modality MAE(Gong et al., 2022; Bachmann et al., 2022) has drawn attention to learning multiple modalities complementary representations. Yet, research on multi-modality masked autoencoders for point cloud learning remains limited. Few Recent frameworks, such as Joint-MAE (Guo et al., 2023) and PiMAE (Chen et al., 2023), propose 2D-3D MAE approaches by reconstructing 2D and 3D inputs. However, we argue that utilizing 2D input is redundant, as 3D data already includes multi-view 2D information. Unlike previous works that require both 3D point clouds and corresponding images for 3D representation learning, Multiview-ML uses only 3D point clouds as input, fully leveraging multi-view information to enhance learning efficiency.

**Multimodal Feature Learning**. Recent studies (Xue et al., 2023; Morgado et al., 2021; Radford et al., 2021; Afham et al., 2022; Jing et al., 2020) have demonstrated that pre-trained multimodal models offer highly transferable representations, significantly boosting performance across various downstream tasks. For example, CrossPoint (Afham et al., 2022) employs contrastive learning to align point clouds with their corresponding 2D images in a shared latent space. MVTN (Hamdi et al., 2021a) enhances 3D object understanding by exploiting multi-view image correspondences. Sautier et al.(Sautier et al., 2022) introduce an object-level contrastive loss between 2D and 3D representations, while TAP(Wang et al., 2023) presents a 3D-to-2D generative pre-training strategy. Unlike prior one-stage approaches that focus solely on reconstructing point clouds, our two-stage strategy emphasizes better alignment between 2D and 3D representations. In this work, we propose a novel two-stage pre-training framework that enhances the alignment of 2D and 3D representations, ensuring that our model learns well-aligned and contextualized multimodal features.

## 3 METHOD

This section introduces our proposed Multiview-ML. In Sec.3.1, we describe how 3D point clouds are projected into multi-view 2D depth maps and encoded with positional, modality, and pose embeddings. Sec.3.2 details our proposed 3D to multi-view encoder architecture, which integrates these multi-view 2D tokens with the original 3D tokens through a fusion encoder. Additionally, Sec.3.3 and Sec.3.4 present the Multi-Scale Multi-Head (MSMH) decoder and our two-stage training strategy, respectively.

### 3.1 3D TO MULTI-VIEW PROJECTION AND ENCODING

**Depth Map Projection.** To establish 2D-3D correspondence, we project the 3D point clouds into multi-view 2D depth images. These depth images then guide the reconstruction from 3D to 2D. We utilize PyTorch3D (Ravi et al., 2020) to project the 3D coordinates onto 2D coordinates using the provided pose information. The 2D coordinates are subsequently converted into token indices using the following formulation:

$$I\hat{d}x = \hat{X}/(H_i//H_t) * H_t + \hat{Y}/(W_i/W_t), \tag{1}$$

where $H_i$, $W_i$ represents the height and width of the depth map. $H_t$, $W_t$ represents the height and width of the 2D token map size. We project 3D coordinates $(X, Y, Z) \in \mathbb{R}^3$ to 2D coordinates $(\hat{X}, \hat{Y}) \in \mathbb{Z}^2$ in a specific view. In our experiments, we use a projected image size of $224 \times 224$ and an image token size of 16, resulting in 196 tokens per image view.

**Positional Encodings.** Following Point-MAE (Pang et al., 2022), our method applies positional encodings to all 3D-related attention layers. For point tokens, we utilize a two-layer `MLP` to encode their corresponding 3D coordinates into $C$-channel vectors $\mathbf{O}^{3D}$, those are then added element-wise to the token features before being fed into the attention layer. Additionally, we add modality-specific 2D sinusoidal positional embeddings $\mathbf{O}^{2D}$ to the image tokens $\mathbf{I}$.

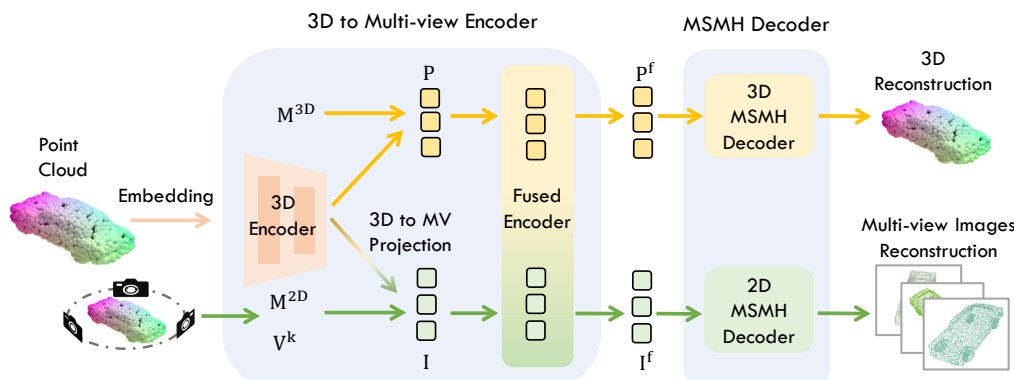

Figure 1: Overview of the stage one training process of the teacher model. The input point clouds are first encoded into intermediate 3D tokens using a 3D encoder. Leveraging the provided pose information, these tokens are projected onto multi-view (MV) 2D image tokens through our proposed view-based feature projection method. Modality and pose embeddings are then added to both the projected 2D tokens and the intermediate 3D tokens. These tokens are concatenated and refined by a multi-modal fusion module. Finally, the refined features are fed into two separate decoders that independently reconstruct the complete multi-view 2D images and the 3D point clouds.

**Modality Encodings.** To enhance the model's ability to distinguish between different modalities within the shared encoder, we add modality-type embeddings $\mathbf{M}^{2D}$ and $\mathbf{M}^{3D}$ to the point tokens $\mathbf{P}$ and image tokens $\mathbf{I}$, respectively, before sending them to the fusion decoder. We employ two-layer `MLP` to encode the modality classes of tokens.

**Pose Encodings.** In the 3D to multi-view autoencoder, our method uses masked 3D tokens to reconstruct fully original depth images from various poses. To incorporate view information into the decoder, we utilize a two-layer `MLP` to encode the view information of each depth map as $\mathbf{V}^k$ for a specific pose $k$ and add it to image tokens before the decoding stage.

**3D to 2D multi-view features projection.** After obtaining the point tokens $\mathbf{P} \in \mathbb{R}^{N_1 \times D}$ from the initial encoder, we project them into the 2D token space of a specific view, as illustrated in Figure 2a. Tokens are then divided into groups $\{\mathbf{T}_i^g\}_{i=1}^G$ according to the index $I\hat{d}x$ obtained from the formula described in Eq. 1. The number of groups $G$ is dynamic, depending on the specific projection. We apply max-pooling and average-pooling to aggregate point tokens that are projected onto the same image tokens. The fused point features are then converted into the image space using an `MLP` layer:

$$\mathbf{I}^g = \mathtt{MLP}\left(\{\mathtt{Max}(\mathbf{T}_i^g) + \mathtt{Ave}(\mathbf{T}_i^g)\}_{i=1}^G\right) \tag{2}$$

As illustrated in Fig. 2a, $\mathbf{I}^g$ cannot be directly mapped to every corresponding 2D view token, leaving certain regions of the projected 2D images blank. Hence, when the number of $\mathbf{I}^g$ is smaller than the 2D image token size (196 in our case), we apply padding for alignment. Subsequently, another `MLP` layer projects $\mathbf{I}^g$ into the specific view $k$, incorporating modality and pose embeddings:

$$\mathbf{I}_k^f = \mathtt{MLP}(\mathtt{Pad}(\mathbf{I}^g)) + \mathbf{M}^{2D} + \mathbf{V}^k + \mathbf{O}^{2D}). \tag{3}$$

### 3.2 3D TO MULTI-VIEW ENCODER

To ensure a fair comparison with previous methods (Guo et al., 2023; Pang et al., 2022), *make our 3D-to-multi-view encoder identical to the plain transformer by adopting the same architecture and equally partitioning its layers into two parts.* The first part functions as the 3D encoder $\mathcal{E}$, responsible for extracting 3D token embeddings, while the remaining layers serve as the multi-modal fusion encoder $\mathcal{E}_r$. Initially, token features $\mathbf{P}$ are extracted from input point clouds using $\mathcal{E}$. These point cloud token features are then mapped to a multi-view image feature space through `MLP`

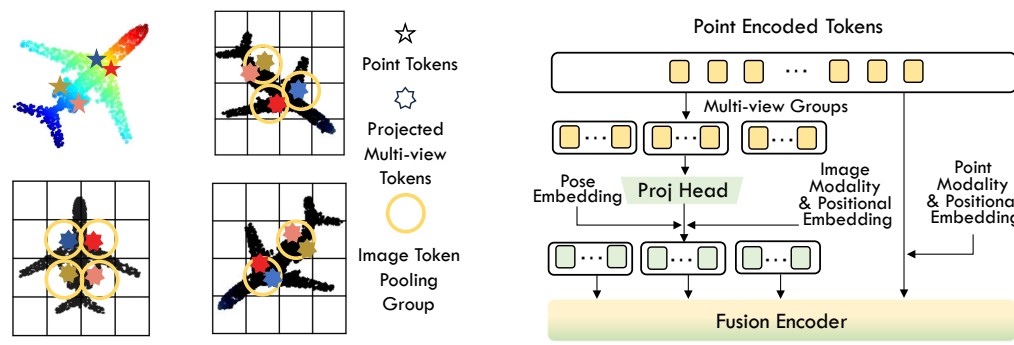

(a) 3D to multi-view tokens-level projection.   (b) Model architecture of 3D to multi-view encoder.

Figure 2: (a) Feature-level projection of 3D point tokens to multi-view 2D tokens. Those point tokens are grouped based on their corresponding image token indices, forming aggregated image token features. (b) The 3D to multi-view encoder for Multiview-ML. Encoded point tokens obtained from 3D encoder are processed by projection heads, projected into multi-view image token positions, and padded to full image token sizes. Those image tokens are concatenated with point tokens and sent to the fusion encoder to further extract both 2D and 3D information.

layers, incorporating 2D modality embeddings and corresponding pose embeddings. This process generates multi-view image tokens $\{\mathbf{I}_k^f\}_{k=1}^K$ for each specific view $k$ as detailed in the Eq. 3. As illustrated in Fig. 2b, we concatenate the multi-view image features $\{\mathbf{I}_k^f\}_{k=1}^K$ with point features $\mathbf{P}^f$ to form a unified embedding. This unified embedding is subsequently processed by the fusion encoder $\mathcal{E}_f$ as follows to obtain refined output $\mathbf{P}^e$ and $\{\mathbf{I}_k^e\}_{k=1}^K$:

$$\mathbf{P}^e, \{\mathbf{I}_i^e\}_{k=1}^K = \mathcal{E}_f\left(\texttt{Concat}(\mathbf{P}^f, \{\mathbf{I}_k^f\}_{k=1}^K)\right), \tag{4}$$

here, $K$ represents the number of views for depth map projection and reconstruction. It is worth noting that even if we refer to the shared encoder and the fusion encoder as different components, they are still the same as the original baseline encoder.

### 3.3 Decoder with Multi-Scale Multi-Head Attention

To effectively capture both local intricacies and global contexts, we introduce an advanced Multi-Scale Multi-Head (MSMH) attention mechanism. This module enables each attention head to independently perform self-attention across various local scopes within point clouds and image segments. Specifically, MSMH attention organizes tokens into distinct, non-overlapping local groups and applies self-attention within each subgroup rather than across all individual tokens. Additionally, our multi-scale design incorporates varying group sizes, allowing smaller groups to capture fine-grained local details while larger groups capture broader global context. We then concatenate the multi-scale attention features to ensure the model integrates both local and global information effectively. By enhancing the baseline model's decoder, we replace the conventional Multi-Head Attention mechanism with our proposed MSMH attention, facilitating the nuanced capture of both local and global attention dynamics in every decoder block.

More formally, the MSMH module is structured with $h$ parallel multi-scale heads, mathematically represented as:

$$\text{MSMH}(\mathbf{Q}, \mathbf{K}, \mathbf{V}) = \texttt{Concat}(\text{ScaleHead}_1, \ldots, \text{ScaleHead}_h)W^O \tag{5}$$

Unlike conventional Multi-Head Attention (MHA) approaches, each head$_i$ in our model performs self-attention within distinct, non-overlapping local scales of size scale$_i$. This is achieved by partitioning the input matrices $\mathbf{Q}\mathbf{W}_i^Q, \mathbf{K}\mathbf{W}_i^K, \mathbf{V}\mathbf{W}_i^V \in \mathbb{R}^{n \times C}$ into smaller segments

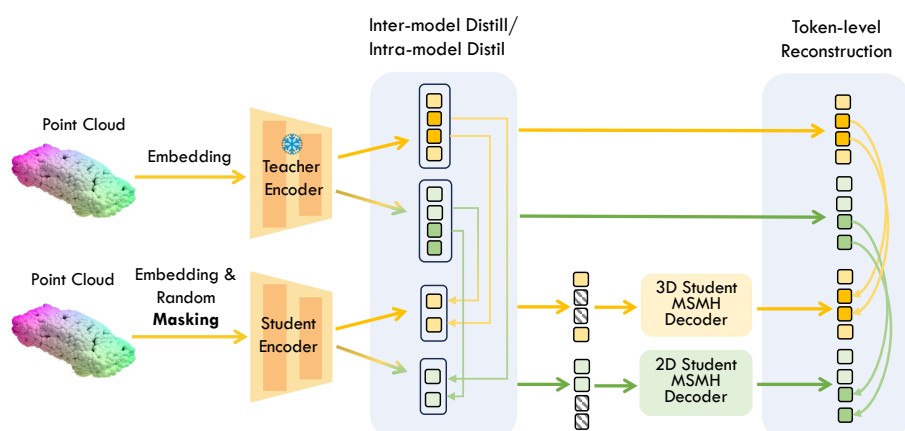

Figure 3: Illustration of the second-stage training process. In this stage, the pre-trained *3D to multi-view encoder* from the first stage act as teacher models, guiding the learning of the student model, which consists of a 3D to multi-view encoder and two MSMH decoders. The training involves two key objectives: (1) Inter/intra model distillation: Global features, obtained by max-pooling from both the teacher and student models, are distilled to transfer global information from the teacher model and ensure feature alignment. (2) Token-level reconstruction: Outputs of the teacher encoder are aligned with the student decoder's predictions for masked tokens feature prediction, facilitating the learning of well-aligned, contextualized representations across both 2D and 3D modalities.

$\mathbf{Q}_{\text{scale}_i}, \mathbf{K}_{\text{scale}_i}, \mathbf{V}_{\text{scale}_i} \in \mathbb{R}^{m \times \text{scale}_i \times C_k}$, where $n = m \times \text{scale}_i$.

$$\text{ScaleHead}_i = \texttt{ScaleAttention}(\mathbf{QW}_i^Q, \mathbf{KW}_i^K, \mathbf{VW}_i^V, \text{scale}_i). \tag{6}$$

The pseudo-code of the ScaleAttention is provided in the Appendix. This is followed by a standard self-attention process: $\mathbf{H}_{scale_i} = \texttt{Attention}(\mathbf{Q}_{scale_i}, \mathbf{K}_{scale_i}, \mathbf{V}_{scale_i})$ on these partitioned inputs. Finally, $\mathbf{H}_{scale_i} \in \mathbb{R}^{m \times scale_i \times C_k}$ is reshaped to $\mathbf{H} \in \mathbb{R}^{n \times C_k}$ to get the output.

In the proposed MSMH module, individual attention heads capture information at multiple local contexts, and the final projection matrix determines the weighting of each head, facilitating interactions across scales. By allowing each head to focus on varying spatial dimensions, the MSMH mechanism facilitates a more dynamic and flexible interpretation of data, effectively capturing both intricate details and broad patterns. This innovative approach significantly enhances the modeling of local and global relationships.

### 3.4 MULTI-MODALITY MASKED FEATURE PREDICTION

Our model, as depicted in Fig. 3, adopts a dual-branch strategy. In this setup, each branch processes different types of inputs: the teacher branch processes unmasked, complete point cloud data, while the student branch deals with the corresponding masked data. During training, the flow of gradients through the teacher branch is intentionally stopped, encouraging the student branch to closely mimic the representations provided by the teacher.

Specifically, the teacher branch's encoders take in unmasked point cloud data to produce aligned and context-rich target representations. The student model is then tasked with reconstructing these target 2D and 3D representations using only the visible (unmasked) 3D inputs. The reconstruction loss is quantified as follows:

$$\mathcal{L}_{token} = \frac{1}{n} \sum_{i=1}^{n} \texttt{MSE}(\mathbf{P}_i^{pre}, \mathbf{P}_i^{teacher}) + \frac{1}{km} \sum_{i=1}^{m} \sum_{k=1}^{K} \texttt{MSE}(\mathbf{I}_{ik}^{pre}, \mathbf{I}_{ik}^{teacher}). \tag{7}$$

Furthermore, we introduce instance-level intra- and inter-modality losses. Our approach shares similarities with SimSiam Chen & He (2021) in encouraging representations from one view to predict

| Method | Foundation model | ScanObjectNN (Uy et al., 2019b) | | | ModelNet40 (Wu et al., 2015) | |
|---|---|---|---|---|---|---|
| | Needed | OBJ-BG | OBJ-ONLY | PB-T50-RS | w/o Vote | w Vote |
| Point-MAE (Pang et al., 2022) | × | 90.02 | 88.29 | 85.18 | 93.2 | 93.8 |
| Joint-MAE (Guo et al., 2023) | × | 90.94 | 88.86 | 86.07 | – | 94.0 |
| TAP (Wang et al., 2023) | × | 90.36 | 89.50 | 85.67 | – | – |
| Point-GPT (Chen et al., 2024) | × | 91.6 | 90.0 | 86.9 | 94.0 | 94.2 |
| **Ours (Point-MAE)** | × | **93.32** (3.3 ↑) | **92.69** (4.4 ↑) | **88.93** (3.8 ↑) | **93.8** (0.6 ↑) | **94.1** (0.3 ↑) |
| Point-M2AE (Zhang et al., 2022) | × | 91.22 | 88.81 | 86.43 | 93.4 | 94.0 |
| **Ours (Point-M2AE)** | × | **95.10** (↑ 3.9) | **93.56** (4.8 ↑) | **90.37** (3.9 ↑) | **94.0** (0.6 ↑) | **94.4** (0.4 ↑) |
| ACT (Dong et al., 2022) | ✓ | 93.29 | 91.91 | 88.21 | 93.7 | – |
| I2P-MAE (Zhang et al., 2023) | ✓ | 94.15 | 91.57 | 90.11 | 93.7 | 94.1 |
| ReCon (Qi et al., 2023) | ✓ | **95.18** | **93.63** | **90.63** | **94.1** | **94.5** |

Table 1: Shape classification performance on ScanObjectNN and ModelNet40, measured by accuracy (%). [Key: **Best results**].

those from another. However, our method extends SimSiam to a multi-modal framework tailored for masked autoencoder networks. Specifically, we pool the representations from visible input tokens and use `MLP` layers to predict both 2D and 3D instance-level representations derived from the complete inputs. This design enhances the learning of meaningful representations across modalities. The intra- and inter-modality distill losses are formulated as:

$$\mathcal{L}_{intra} = \texttt{MSE}(\texttt{MLP}(\mathbf{P}_{ins}^{pre}), \mathbf{P}_{ins}^{teacher}) + \texttt{MSE}(\texttt{MLP}(\mathbf{I}_{ins}^{pre}), \mathbf{I}_{ins}^{teacher}) \qquad (8)$$

$$\mathcal{L}_{inter} = \texttt{MSE}(\texttt{MLP}(\mathbf{P}_{ins}^{pre}), \mathbf{I}_{ins}^{teacher}) + \texttt{MSE}(\texttt{MLP}(\mathbf{I}_{ins}^{pre}), \mathbf{P}_{ins}^{teacher}) \qquad (9)$$

Finally, we sum up all three losses as the final loss for our method:

$$\mathcal{L}_{final} = 0.5\mathcal{L}_{intra} + 0.5\mathcal{L}_{inter} + \mathcal{L}_{token} \qquad (10)$$

In the first stage of training, the teacher model is trained on a raw input reconstruction task using full point cloud data without masking. The objective is to reconstruct both the complete point clouds and the corresponding multi-view depth images. Loss functions for this phase include Chamfer Distance for 3D coordinates and Mean Squared Error (MSE) for the 2D depth images across multiple views:

$$\mathcal{L}_{teacher} = \frac{1}{n}\sum_{i=1}^{N}\texttt{Chamfer}(\mathbf{P}_i^{pre}, \mathbf{P}_i^{gt}) + \frac{1}{km}\sum_{i=1}^{k}\sum_{j=1}^{M}\texttt{MSE}(\mathbf{I}_{ij}^{pre}, \mathbf{I}_{ij}^{gt}). \qquad (11)$$

After pre-training the teacher model, we leverage the pre-trained teacher from the first stage and keep it frozen during the main training phase. This two-stage design ensures that the student models learn well-aligned and contextualized representations across modalities.

# 4 EXPERIMENTS

## 4.1 SELF-SUPERVISED PRE-TRAINING

**Datasets.** In our experiments, we use several datasets, including ShapeNet (Chang et al., 2015), ModelNet40 (Wu et al., 2015), ScanObjectNN (Uy et al., 2019b), ShapeNetPart dataset (Yi et al., 2016), and ScanNetV2 dataset (Dai et al., 2017). The ShapeNet (Chang et al., 2015) comprises about $51,300$ clean 3D models. The widely adopted ModelNet40 (Wu et al., 2015) consists of synthetic 3D shapes of 40 categories, of which $9,843$ samples are for training and the other $2,468$ are for validation. The challenging ScanObjectNN (Uy et al., 2019a) contains $11,416$ training and $2,882$ validation point clouds of 15 categories. ShapeNet (Chang et al., 2015) dataset. ScanObjectNN is divided into three splits for evaluation, OBJ-BG, OBJ-ONLY, and PB-T50-RS. ShapeNetPart (Yi et al., 2016) is a widely used dataset for 3D semantic segmentation, which consists of $16,881$ models

| Method | Foundation Model | 5-way | | 10-way | |
|---|---|---|---|---|---|
| | Needed | 10-shot | 20-shot | 10-shot | 20-shot |
| Point-BERT (Yu et al., 2021) | × | 94.6 ± 3.1 | 96.3 ± 2.7 | 91.0 ± 5.4 | 92.7 ± 5.1 |
| Point-MAE (Pang et al., 2022) | × | 96.3 ± 2.5 | 97.8 ± 1.8 | 92.6 ± 4.1 | 95.0 ± 3.0 |
| Joint-MAE (Guo et al., 2023) | × | 96.7 ± 2.2 | 97.9 ± 1.8 | 92.6 ± 3.7 | 95.1 ± 2.6 |
| Point-M2AE (Zhang et al., 2022) | × | 96.8 ± 1.8 | 98.3 ± 1.4 | 92.6 ± 5.0 | 95.0 ± 3.0 |
| TAP (Wang et al., 2023) | × | 97.3 ± 1.8 | 97.8 ± 1.7 | 93.1 ± 2.6 | 95.8 ± 1.0 |
| Ours(Point-MAE) | × | 97.3 ± 1.9 | 98.2 ± 1.6 | 93.2 ± 4.1 | 96.0 ± 2.7 |
| **Ours (Point-M2AE)** | × | **97.6 ± 2.1** | **98.5 ± 1.3** | **93.6 ± 3.9** | **96.1 ± 2.1** |
| ACT (Dong et al., 2022) | ✓ | 96.8 ± 2.3 | 98.0 ± 1.4 | 93.3 ± 4.0 | 95.6 ± 2.8 |
| I2P-MAE (Zhang et al., 2023) | ✓ | 97.0 ± 1.8 | 98.3 ± 1.4 | 92.3 ± 4.5 | 95.5 ± 3.0 |
| ReCon (Qi et al., 2023) | ✓ | **97.3 ± 1.9** | **98.9 ± 1.2** | **93.3 ± 3.9** | **95.8 ± 3.0** |

Table 2: Few-shot classification performance on ModelNet40 (Wu et al., 2015), measured by the accuracy (%) and standard deviation (%). ∗ denotes the model without pre-training. [Key: **Best results**, Second best results.].

across 16 categories. The ScanNet (Dai et al., 2017) is an indoor scene dataset consisting of $1,513$ reconstructed meshes, among which $1,201$ are training samples and 312 are validation samples.

**Settings.** We utilize the ShapeNet (Chang et al., 2015) for pre-training. To obtain a dense depth map, the input point number $N$ is set as the baseline method. The number of the projection view $K$ is set to 3 and the depth map size is set as 224×224. Random scaling and random rotation are implemented as data augmentation during pre-training. We project point clouds into multi-view after the augmentation. Our method employs an AdamW optimizer (Loshchilov & Hutter, 2017) and cosine learning rate decay (Loshchilov & Hutter, 2016). The network is trained for 300 epochs with a batch size of 128. The initial learning rate, weight decay, and mask ratio are set to $2 \times 10^{-4}$, 0.05, and 0.7. The scale for 3D and 2D modality is $[2, 4, 8, 16, 32, 64]$ and $[6, 12, 24, 49, 98, 196]$.

## 4.2 DOWNSTREAM TASKS

**Shape Classification.** We fine-tune the proposed method on ModelNet40 (Wu et al., 2015) and ScanObjectNN (Uy et al., 2019b) datasets, which contain synthetic objects and real-world instances, respectively. During training in both dataset, we utilize data augmentation techniques such as random scaling and random translation. For the ModelNet40 dataset, we use the standard voting method (Liu et al., 2019) as Point-BERT for a fair comparison. As shown in Table 1, in the ModelNet40 dataset, our method achieves an accuracy of 94.4%, which is an improvement of 3.0% compared to training from scratch (91.4%) and surpasses the baseline Point-M2AE by 0.4%. This improvement is significant, considering that ModelNet40 is a relatively small dataset. For the ScanObjectNN dataset, we conduct experiments on three variants: OBJ-BG, OBJ-ONLY, and PBT50-RS. As shown in Table 1, our method significantly improves the baseline Point-MAE and Point-M2AE by a large margin. It is important to highlight that our performance also significantly exceeds that of Joint-MAE (Guo et al., 2023), which employs a multi-modality MAE structure but overlooks the intrinsic multi-view nature of point clouds. This underscores the critical importance of harnessing multi-view information in 3D pre-training tasks, demonstrating that exploiting rich perceptual features derived from multi-view projections of point clouds benefits the 3D representation learning and thus can yield considerable improvements in downstream tasks. It is noteworthy that our method outperforms approaches distilled from foundation models, such as ACT Dong et al. (2022) and I2P-MAE Zhang et al. (2023), and achieves results comparable to the SOTA ReCon Qi et al. (2023). However, it is important to note that ReCon modifies the fine-tuning stages of the baseline Point-MAE, significantly enhancing its performance. As a result, a direct comparison between our method and ReCon is inherently unfair.

**Few-shot Learning.** We conducted few-shot learning experiments on the ModelNet40 dataset (Wu et al., 2015) using the $n$-way, $m$-shot setting, following the protocol of Point-MAE. During training, we randomly selected $n$ classes and $m$ objects from each class. During testing, we randomly selected 20 unseen objects from each of the $n$ classes for evaluation. We conducted 10 independent experiments for each setting and reported the mean accuracy with standard deviation. The results of

| Method | [P] | mIoU$_C$ | mIoU$_I$ |
|---|---|---|---|
| PointNet (Qi et al., 2017a) | | 80.39 | 83.70 |
| Transformer (Yu et al., 2021) | | 83.42 | 85.10 |
| Point-BERT (Yu et al., 2021) | ✓ | 84.11 | 85.60 |
| Point-MAE (Pang et al., 2022) | ✓ | - | 86.10 |
| Joint-MAE (Pang et al., 2022) | ✓ | 85.41 | 86.28 |
| Point-M2AE (Zhang et al., 2022) | ✓ | 84.86 | 86.51 |
| ACT (Dong et al., 2022) | ✓ | 84.66 | 86.14 |
| I2P-MAE (Zhang et al., 2023) | ✓ | 85.15 | 86.76 |
| ReCon (Qi et al., 2023) | ✓ | 84.80 | 86.40 |
| Ours(Point-MAE) | ✓ | 85.58 | 86.79 |
| **Ours(Point-M2AE)** | ✓ | **85.66** | **86.91** |

Table 3: Part segmentation on ShapeNetPart (Yi et al., 2016). mIoU$_C$ (%) and mIoU$_I$ (%) denote the mean IoU across all part categories and all instances in the dataset, respectively. [P] represents fine-tuning after self-supervised pre-training.

| Methods | [P] | $AP_{25}$ | $AP_{50}$ |
|---|---|---|---|
| VoteNet (Qi et al., 2019) | | 58.6 | 33.5 |
| PointContrast (Xie et al., 2020) | ✓ | 59.2 | 38.0 |
| DepthContrast (Zhang et al., 2021) | ✓ | 64.0 | 42.9 |
| DPCo (Li & Heizmann, 2022) | ✓ | 64.2 | 41.5 |
| 3DETR (Misra et al., 2021) | | 62.1 | 37.9 |
| +Point-BERT(Yu et al., 2022) | ✓ | 61.0 | 38.3 |
| +Point-MAE (Pang et al., 2022) | ✓ | 62.8 | 40.1 |
| +MaskPoint (Liu et al., 2022) | ✓ | 63.4 | 40.6 |
| +ACT (Dong et al., 2022) | ✓ | 63.5 | 41.0 |
| +PiMAE (Chen et al., 2023) | ✓ | 63.1 | 40.8 |
| +TAP (Wang et al., 2023) | ✓ | 63.0 | 41.4 |
| **+ Ours** | ✓ | **63.9** | **43.3** |

Table 4: 3D object detection results on ScanNet dataset. We adopt the average precision with 3D IoU thresholds of 0.25 ($AP_{25}$) and 0.5 ($AP_{50}$) for the evaluation metrics. [P] represents fine-tuning after self-supervised pre-training.

our fine-tuned few-shot classification are shown in Table 2. Our method outperformed the baseline and state-of-the-art methods in all settings.

**Part Segmentation.** We evaluate our method's representation learning capability on the ShapeNet-Part dataset (Yi et al., 2016), which contains 14,007 and 2,874 samples with 16 object categories and 50 part categories for training and validation. Following previous method (Pang et al., 2022), we sample 2,048 points from each input instance and adopt the same segmentation head (Qi et al., 2017a;b) for the fair comparison, which concatenates the output features from different transformer blocks of the encoder. The head only conducts simple upsampling for point tokens at different stages and concatenates them alone with the feature dimension as the output. We report mean IoU (mIoU) for all instances, with IoU for each category. Table 3 indicates that our method improves the baseline method Point-MAE and surpasses the state-of-the-art method Point-M2AE in all settings.

**3D Object Detection.** To demonstrate the generality of the proposed method, we also pre-train the Multiview-ML on the indoor ScanNetV2 dataset (Dai et al., 2017) and subsequently fine-tune our method on the object detection task. Our baseline is 3DETR (Misra et al., 2021), which consists of a 3-block encoder and a transformer decoder. We utilize the same backbone as 3DETR. The Table 4 indicates that Our method achieves 63.9 AP$_{25}$ (+1.8) and 43.3 AP$_{50}$ (+5.4) compared to the baseline 3DETR on the ScanNetV2 (Dai et al., 2017) dataset. Furthermore, we benchmark our method against PiMAE (Chen et al., 2023), a novel approach introduced in the CVPR 2023 paper, which integrates a multi-modality structure. PiMAE is designed to leverage both masked single RGB images and point clouds for reconstructing the original image and point clouds. Despite PiMAE's ability to enable point clouds to learn texture information from images, it overlooks the multi-view characteristics of point clouds. Our method's considerable superiority over PiMAE underscores the critical importance of leveraging multi-view information for effective 3D representation learning.

## 4.3 ABLATION STUDY

**The Effectiveness of Each Proposed Component.** The ablation study presented in Table 5 underscores the substantial impact of our proposed 3D to multi-view Multi-Modal Learner in advancing 3D representation learning, particularly in comparison to the baseline Point-MAE (Pang et al., 2022), which relies solely on the 3D modality. Our method, which directly reconstructs multi-modality masked representations from visible point clouds, already demonstrates a significant performance boost over Point-MAE. This observation underscores the efficacy of token-level representation prediction for 3D representation learning. Moreover, our approach leverages instance-level intra and inter-modality representation prediction, further enhancing performance. By utilizing global features of masked tokens to predict global representations derived from complete inputs, the network gains a deeper understanding of global information and becomes more robust, thus improving representation learning. Crucially, with the incorporation of the proposed Multi-Scale Multi-Head Attention in the decoder, our framework achieves its peak performance. This underscores the effectiveness

| Toekn Rec | Instance Rec | MSMH | PB-T50-RS |
|:---:|:---:|:---:|:---:|
| - | - | - | 85.18 |
| ✓ | - | - | 87.07 |
| - | ✓ | - | 86.26 |
| - | - | ✓ | 86.03 |
| ✓ | ✓ | - | 88.11 |
| ✓ | - | ✓ | 87.52 |
| - | ✓ | ✓ | 87.19 |
| ✓ | ✓ | ✓ | **88.93** |

Table 5: Ablation study for the effectiveness of each proposed component on the 3D classification task in ScanObjectNN dataset.

| Input Modality | Output Modality | PB-T50-RS |
|:---:|:---:|:---:|
| 3D | 3D | 86.76 |
| 3D | 2D | 84.25 |
| 3D & 2D | 3D & 2D | 87.02 |
| 3D | 3D & 2D | **88.93** |

Table 6: Ablation study for the effectiveness of input and output modalities on the 3D object classification task in ScanObjectNN dataset.

of Multi-Scale Multi-Head Attention in capturing both local and global information comprehensively. This attention mechanism facilitates the model in effectively learning intricate patterns at various scales, leading to a superior representation of learning outcomes.

**The Effectiveness of Input and Output Modality.** The Effectiveness of Input and Output Modality. We introduce a novel method to reconstruct both 3D and 2D multi-view representations using only 3D data. Our ablation study (Table 6) demonstrates the effectiveness of the 3D to multi-view feature projection technique. By maintaining consistent settings for masked token reconstruction, intra/inter-level representation predictions, and Multi-Scale Multi-Head Attention, our approach with solely 3D input and output significantly outperforms the baseline Point-MAE, highlighting its strength in 3D representation learning. Incorporating both 3D and 2D inputs for reconstructing 3D and 2D outputs yields only marginal gains, similar to findings in Joint-MAE (Guo et al., 2023). Notably, our method, which *exclusively uses 3D inputs*, demonstrated the best overall performance. We argue that adding 2D depth leads the network to heavily rely on the visible 2D information to predict the masked 2D content without the need to fully understand the multi-view geometry setting and thus degrade the representation learning. This finding illustrates the added value of integrating multi-view and pose-related information into the 3D representation learning paradigm

**The Effectiveness of the Two-stage Framework.** As discussed in previous work (Chen & He, 2021), latent space prediction is highly effective for SSL representation learning. Inspired by this, we introduce latent space prediction into the multi-modality MAE in this paper. Rather than predicting raw inputs across diverse modalities, our model uses masked-view inputs

| Method | OBJ-BG | OBJ-ONLY | PB-T50-RS |
|:---|:---:|:---:|:---:|
| Stage 1 + MAE | 92.34 | 91.88 | 87.56 |
| Ours | **93.32** | **92.69** | **88.93** |

Table 7: Ablation study for the two-stage design.

to jointly predict contextualized 2D and 3D representations within a latent space aligned by a teacher model with complete-view inputs. This ensures that student models learn well-aligned and contextualized representations across modalities. Our ablation study, shown in Table 7, demonstrates the advantage of this two-stage latent prediction method. Compared to the direct reconstruction of masked multi-modality raw inputs in one stage, our approach achieves 1.37 % improvement.

## 5 CONCLUSION

This paper introduces Multiview-ML, an innovative technique for 3D representation learning that capitalizes on the multi-view nature of 3D point clouds. Unlike conventional approaches, our method uniquely employs masked point clouds to reconstruct their original forms and generate multiple depth images from different views. This approach harnesses the rich, multi-view features and spatial information within point clouds, distinguishing our method from existing multi-modal MAE techniques. Additionally, we introduce a multi-scale multi-head attention mechanism that enhances the interplay between local and global perspectives within each decoder transformer block, utilizing attention heads across various scales. Finally, a novel self-training strategy is proposed, aiming to generate aligned and context-rich masked 2D and 3D representations based on the initial learning objectives. Our experimental results showcase the superior performance of Multiview-ML over current leading self-supervised 3D learning methods across various downstream applications, highlighting its effectiveness in fully leveraging the multi-view attributes of 3D data.

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
