# Supplementary Material for Point Cloud Self-supervised Learning via 3D to Multi-view Masked Learner

| Method | ScanObjectNN (Uy et al., 2019b) | | | ModelNet40 (Wu et al., 2015) | |
|---|---|---|---|---|---|
| | OBJ-BG | OBJ-ONLY | PB-T50-RS | w/o Vote | w/ Vote |
| I2P-MAE (Zhang et al., 2023) | 94.15 | 91.57 | 90.11 | 93.7 | 94.1 |
| **Ours (I2P-MAE)** | **95.72** (↑ 1.57) | **94.28** (2.71 ↑) | **91.36** (1.25 ↑) | **94.0** (0.3 ↑) | **94.3** (0.2 ↑) |
| ReCon (Qi et al., 2023) | 95.18 | 93.63 | 90.63 | 94.1 | 94.5 |
| **Ours (ReCon)** | **96.03** (↑ 0.85) | **95.32** (1.69 ↑) | **92.06** (1.43 ↑) | **94.3** (0.2 ↑) | **94.7** (0.2 ↑) |

Table 1: Experiment results of apply our method on I2P-MAE and ReCon. Shape classification performance on ScanObjectNN and ModelNet40, measured by accuracy (%).

## A  Comparison and Compatible with Models Trained via Foundation Models

**Comparison with I2P-MAE (Zhang et al., 2023), ReCon (Qi et al., 2023), and ShapeLLM (Qi et al., 2025).**

I2P-MAE, ReCon, and ShapeLLM focus primarily on two types of features: 3D geometric features and semantic/textual representations. They employ MAE-based structures to reconstruct the original point clouds, thereby capturing detailed 3D geometric data. Additionally, they utilize techniques such as contrastive learning or knowledge distillation to extract semantic and textual features from 2D images and language models. These methods directly adopt existing 3D MAE frameworks—specifically, I2P-MAE utilizes Point-M2AE, while ReCon and ShapeLLM leverage Point-MAE for geometric representation—and their innovation lies in the novel use of foundation models for knowledge distillation.

I2P-MAE performs pixel-to-3D token knowledge distillation by adding additional layers after the M2AE encoder, calculating MSE loss between the point tokens and 2D pixel-level features derived from foundation models. ReCon uses Point-MAE as the base structure to reconstruct original point clouds from masked point cloud inputs, while also incorporating instance-level contrastive learning to distill knowledge from both text and image foundation models. ShapeLLM builds upon ReCon by using larger models with more parameters, leveraging large language models to enable advanced 3D reasoning. **In contrast**, our approach focuses on advancing geometric learning in 3D self-supervised learning (SSL), *emphasizing the use of the inherent multi-view attributes in point cloud data to enhance geometric understanding*, solely within the 3D modality. Due to the fundamental differences in goals and methodologies, a direct comparison with I2P-MAE and ReCon would not provide a fair evaluation.

**Compatible with I2P-MAE (Zhang et al., 2023) and ReCon (Qi et al., 2023).** While I2P-MAE and ReCon primarily focus on leveraging foundation models for knowledge distillation, our method centers on advancing 3D geometric learning. To demonstrate the generality of our approach, we integrate it into both I2P-MAE and ReCon, enhancing their ability to capture 3D geometric information. Specifically, we incorporate our proposed 3D-to-multi-view projection into the original encoder of I2P-MAE and ReCon and introduce the MSMH module to enable the reconstruction of multi-view images solely from 3D input. Experimental results indicate that our approach improves the performance of these foundation model distillation methods. We attribute this improvement to

the fact that foundation models like CLIP are primarily trained to capture semantic information but lack a deep understanding of 3D geometric structures. In contrast, our method leverages the inherent multi-modality of point cloud data to enhance geometric understanding, making it complementary to foundation models and improving 3D representation learning.

## B    COMPARISON WITH 3D GEOMETRIC LEARNING SSL METHODS

Our work focuses on 3D geometric learning without leveraging foundation models, similar to methods like Point-M2AE (Zhang et al., 2022), Point-GPT (Chen et al., 2024), Pi-MAE (Chen et al., 2023), Joint-MAE (Guo et al., 2023), and TAP (Wang et al., 2023), which aim to learn pure 3D geometric representations without relying on knowledge distillation from foundation models. Existing MAE-based 3D geometric learning methods generally follow two modification directions: (1) Encoder structure modification, as seen in methods like Point-M2AE and Point-GPT, and (2) Incorporating 2D information into the reconstruction process, as done by Pi-MAE, Joint-MAE, and TAP.

Our work follows the second direction but addresses significant limitations in existing methods that leverage 2D information for 3D geometric learning. Specifically, approaches like Pi-MAE, Joint-MAE, and TAP do not fully exploit the multi-view properties of 3D point clouds and their inherently multi-modal attributes. For example, a point cloud can be directly projected into multi-view images using pose information. Incorporating masked 2D images as input, as done by Pi-MAE and Joint-MAE during the MAE training process, is unnecessary and potentially detrimental, as it can cause the network to over-rely on visible 2D information to predict masked content rather than developing a comprehensive understanding of multi-view geometry, ultimately degrading the quality of learned 3D representations. Moreover, TAP uses a pre-trained VAE to reconstruct 2D images from 3D inputs but fails to effectively leverage multi-view information. In contrast, our method introduces a unified approach that uses masked point clouds to reconstruct both multi-view 2D images and the original point clouds, ensuring a more comprehensive understanding of 3D geometry while effectively utilizing the multi-view attributes of 3D data. Furthermore, we propose MSMH decoder to better global and local features and a two-stage self-training method to learn well-aligned representations. It is worth to mention that during the fine-tuning and inference stages, we remove additional components, such as the projection layers and MSMH decoder, maintaining the same architecture as Point-MAE to ensure a fair comparison.

## C    ADDITIONAL ABLATION STUDY

**The Effectiveness of Poses Pool Size.** The pose pool size represents the total number of poses that can be leveraged in our 3D to multi-view MAE method. The ablation study detailed in Table 2 investigates the impact of varying the number of views in the network on 3D object classification performance, using the ScanObjectNN dataset. The study examines a range of views: 3, 6, 12, 24, and 36 to understand how they affect classification accuracy. The results reveal a notable trend: as the number of views increases, there's generally an improvement in classification accuracy, achieving the best performance at 12 views. Beyond this optimal point, however, the performance decreases with the increase of projected views. This pattern indicates that while increasing the number of views contributes positively to the network's understanding and representation of 3D objects, there is a point beyond which additional views do not yield further benefits. This is because too many views introduce the redundancy of view-specific information, leading to a slight decrease in the network's efficiency.

**Effectiveness of Image Type.** In the ablation study presented in Table 4 , we analyze two commonly used image types for 3D understanding: rendered images and depth images. The results indicate that using depth images yields the best performance, which aligns with findings from previous work, such as Joint-MAE.

**The Effectiveness of Network Reconstructed View Numbers.** Our method enhances multi-view understanding by randomly selecting several view poses from the pose pool mentioned above, enabling the model to reconstruct corresponding multiple projected depth images. This ablation study focuses on finding the optimal number of reconstructed views for enhancing 3D representation learning in the ScanObjectNN (Uy et al., 2019b) dataset. We examined the impact of the number of re-

| # Pose Pool Size | PB-T50-RS |
|:---:|:---:|
| 3 | 87.17 |
| 6 | 88.29 |
| 12 | **88.93** |
| 24 | 88.54 |
| 36 | 87.76 |

Table 2: Ablation study for the number of pose pool size on the 3D object classification tasks in ScanObjectNN dataset.

| # Recon View Size | PB-T50-RS |
|:---:|:---:|
| 1 | 87.05 |
| 2 | 88.41 |
| 3 | **88.93** |
| 4 | 88.12 |
| 5 | 87.58 |

Table 3: Ablation study for the number of reconstructed views on the 3D object classification task in ScanObjectNN dataset.

| Image Type | PB-T50-RS |
|:---:|:---:|
| Depth Image | **88.93** |
| Rendered Image | 88.12 |

Table 4: Ablation study for the image type on the 3D object classification tasks in ScanObjectNN dataset.

| View Configuration | PB-T50-RS |
|:---:|:---:|
| Circular | **88.93** |
| Spheric | 87.97 |
| Spheric & Circular | 88.35 |
| Random | 87.41 |

Table 5: Ablation study for the view configuration of the depth images on the 3D object classification tasks in ScanObjectNN dataset.

constructed views from one to five on classification performance in PB-T50-RS setting. According to the results in Table 3, accuracy consistently increases with the number of views, peaking at 3 views. Beyond this point, however, the trend indicates a decrease in performance. This suggests that multiple reconstructed views enhance the network's understanding of multi-view information. However, too many reconstructed views will make the length of the input sequences processed by the decoder very large, thus impacting the network's learning efficiency and capacity.

**The Effectiveness of View Configurations.** In the ablation study shown in Table 5, different view configurations of depth images for our method in 3D representation learning are analyzed using the ScanObjectNN dataset. The most common view configurations for depth image projection are circular which alignes viewpoints on a circle around the object (Su et al., 2015; Yu et al., 2018) and spherical which alignes equally spaced viewpoints on a sphere surrounding the object (Wei et al., 2020; Kanezaki et al., 2018). We test Circular, Spheric, a combination of both, and Random configurations. The Circular configuration proves most effective, achieving the highest accuracies of 88.93 in PB-T50-R, likely due to its comprehensive coverage and consistent viewing angles. The Spheric configuration, while offering a broad perspective, falls slightly short in comparison. Combining Spheric and Circular views improves performance but does not outperform the Circular configuration alone. The Random configuration shows the least effectiveness. This study highlights the Circular view configuration's superiority in providing a balanced and thorough representation of 3D objects, essential for better representation learning.

**The Effectiveness of Pose Type.** The ablation study detailed in Table 6 critically examines the influence of pose type on the accuracy of 3D object classification within the ScanObjectNN dataset. It delves into two distinct pose types: Index and Camera Matrix, assessing their effectiveness in PB-T50-R setting of the ScanObjectNN dataset. The Index pose type employs fixed indexes to denote specific pose views, whereas the Camera Matrix approach directly inputs the camera matrix into the pose encoding process to derive pose embeddings. Notably, both pose types demonstrate commendable performance, with the Index slightly surpassing the Camera Matrix. This marginal difference underscores the robustness of the classification method to variations in pose type input, suggesting a flexible adaptability to different pose representation strategies in 3D representation learning.

**The Effectiveness of Reconstruction Type.** In this research, we leverage the student branch to reconstruct the representations of masked tokens based on guidance from the teacher branch. Our ablation study, presented in Table 7, meticulously evaluates the influence of various reconstruction (Rec) methodologies on the 3D object classification accuracy using the ScanObjectNN dataset. This study differentiates between three reconstruction types: 'Masked Only', 'Full', and 'Visible Only'.

| Pose Type | PB-T50-RS |
|---|---|
| Index | **88.93** |
| Camera Matrix | 88.33 |

Table 6: Ablation study for the pose type on the 3D object classification tasks in ScanObjectNN dataset.

| Rec Type | PB-T50-RS |
|---|---|
| Masked Only | **88.93** |
| Full | 88.41 |
| Visible Only | 87.74 |

Table 7: Ablation study for the feature reconstruction type on the 3D object classification tasks in ScanObjectNN dataset.

The findings indicate that focusing on reconstructing only the masked features yields the most favorable outcomes. In contrast, the approach of reconstructing only the visible features, similar to the previous state-of-the-art method I2P-MAE, results in the least effective performance. These results underscore the effectiveness of our proposed method in more accurately aligning the latent spaces of the teacher and student models and the better ability to fully utilize the multi-view information.

**The Effectiveness of Masking Ratio.** The ablation study outlined in Table 8 evaluates the effect of different masking ratios on 3D representation learning in the ScanObjectNN dataset. Five masking ratios are tested: 0.6, 0.65, 0.7, 0.75, and 0.8, assessing their impact on performance in PB-T50-RS setting in the ScanObjectNN dataset. The results indicate a clear pattern. As the masking ratio decreases from 0.6 to 0.7, classification accuracy consistently improves. The best performance is observed at a masking ratio of 0.7, with accuracies reaching 88.93%. However, reducing the masking ratio further to 0.8

| Masking Ratio | PB-T50-RS |
|---|---|
| 0.6 | 88.02 |
| 0.65 | 88.46 |
| **0.7** | **88.93** |
| 0.75 | 88.15 |
| 0.8 | 87.62 |

Table 8: Ablation study for the masking ratio on the 3D object classification tasks in ScanObjectNN dataset.

results in a slight decrease in accuracy. These findings suggest that an optimal masking ratio exists, where a balance is struck between challenging the network sufficiently to learn robust features and retaining enough information for accurate classification. Too much masking may obscure critical details, while too little may not provide enough complexity for effective learning.

# D  ADDITIONAL EXPERIMENTS

## D.1  PART SEGMENTATION

As shown in Table 9, we report mean IoU (mIoU) for all instances, with IoU for each category. Our method achieves the best performance in all categories.

## D.2  LINEAR SVM RESULT

To evaluate the transfer capacity, we directly utilize the features extracted by I2P-MAE's encoder for linear SVM on the synthetic ModelNet40 (Wu et al., 2015) without any fine-tuning or voting. The results on ModelNet40 are shown in Table 10. It shows that our RECON outperforms the last SOTA method I2P-MAE (Zhang et al., 2023) by 0.3% even without using pre-trained foundation models. This improvement in SVM classification performance underscores the efficacy of our approach in learning superior quality 3D representations and highlights the value of the inherent multi-view property of 3D data.

# E  VISUALIZATION

For the second-stage design, our method focuses on feature reconstruction. Therefore, visualizing the reconstruction across the entire two-stage process poses significant challenges. To address this, we provide visualization results by directly integrating MAE into the stage-one framework, as detailed in Table 7 of the main paper. The visualization results are presented in Fig. 1, where each row illustrates the input point clouds, masked point clouds, reconstructed point clouds, projected depth

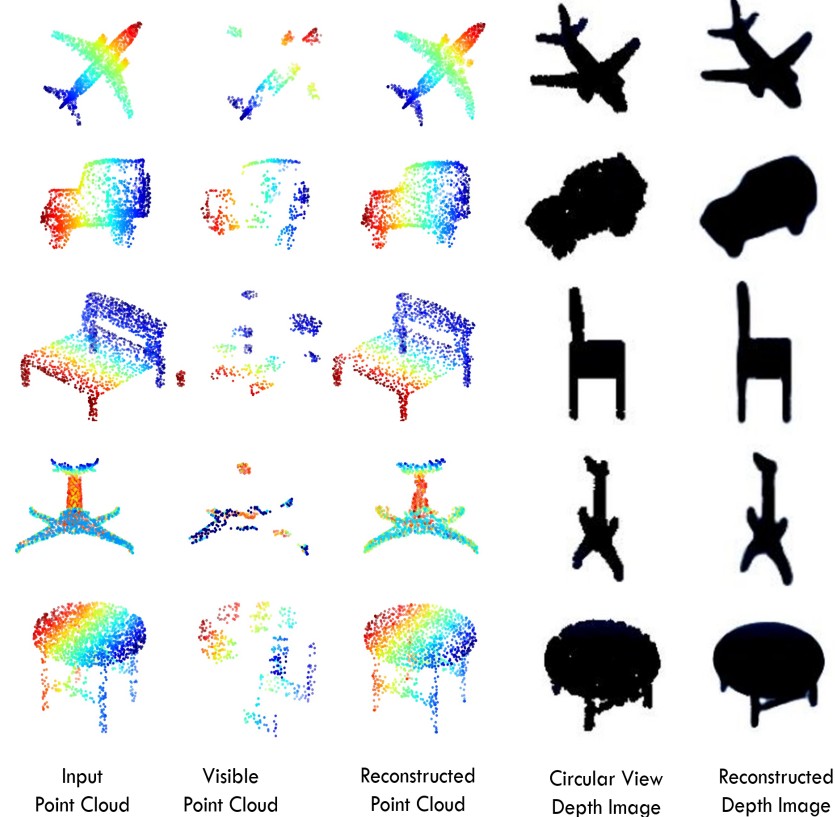

Figure 1: Visualization of 3D to multi-view masked autoencoder (Stage 1 with MAE). Our method not only can reconstruct point clouds from masked input but also generate multi-view depth images.

| Method | mIoU$_C$ | mIoU$_I$ | aero | bag | cap | car | chair | e-phone | guitar | knife | lamp | laptop | motorbike | mug | pistol | rocket | skateboard | table |
|---|---|---|---|---|---|---|---|---|---|---|---|---|---|---|---|---|---|---|
| PointNet Qi et al. (2017a) | 80.39 | 83.7 | 83.4 | 78.7 | 82.5 | 74.9 | 89.6 | 73.0 | 91.5 | 85.9 | 80.8 | 95.3 | 65.2 | 93.0 | 81.2 | 57.9 | 72.8 | 80.6 |
| PointNet++ Qi et al. (2017b) | 81.85 | 85.1 | 82.4 | 79.0 | 87.7 | 77.3 | 90.8 | 71.8 | 91.0 | 85.9 | 83.7 | 95.3 | 71.6 | 94.1 | 81.3 | 58.7 | 76.4 | 82.6 |
| Transformer Yu et al. (2021) | 83.42 | 85.1 | 82.9 | 85.4 | 87.7 | 78.8 | 90.5 | 80.8 | 91.1 | 87.7 | 85.3 | 95.6 | 73.9 | 94.9 | 83.5 | 61.2 | 74.9 | 80.6 |
| Point-BERT Yu et al. (2021) | 84.11 | 85.6 | 84.3 | 84.8 | 88.0 | 79.8 | 91.0 | 81.7 | 91.6 | 87.9 | 85.2 | 95.6 | 75.6 | 94.6 | 84.7 | 63.4 | 76.3 | 81.5 |
| Point-MAE (Pang et al., 2022) | 84.19 | 86.1 | 84.3 | 85.0 | 88.3 | 80.5 | 91.3 | 78.5 | 92.1 | 87.4 | 86.1 | 96.1 | 75.2 | 94.6 | 84.7 | 63.5 | 77.1 | **82.4** |
| **Ours** | **85.66** | **86.9** | **85.1** | **86.0** | **89.3** | **82.7** | **91.4** | **80.5** | **93.4** | **88.7** | **87.4** | **96.8** | **77.1** | **96.1** | **86.3** | **68.8** | **78.5** | **82.4** |

Table 9: Part segmentation on ShapeNetPart (Yi et al., 2016). We report mean IoU for all instances mIoU (%), with IoU (%) for each category.

images, and reconstructed images, respectively. Our method demonstrates the ability to not only reconstruct point clouds from masked inputs but also generate multiview depth images, highlighting its capability to effectively capture the intrinsic multi-modal information of point clouds.

## F DATASETS

In our experiments, we use several datasets, including ShapeNet (Chang et al., 2015), Model-Net40 (Wu et al., 2015), ScanObjectNN (Uy et al., 2019b), ShapeNetPart dataset (Yi et al., 2016), and ScanNetV2 dataset (Dai et al., 2017). The ShapeNet (Chang et al., 2015) comprises about $51,300$ clean 3D models covering 55 common object categories. The widely adopted Model-Net40 (Wu et al., 2015) consists of synthetic 3D shapes of 40 categories, of which $9,843$ samples are for training and the other $2,468$ are for validation. The challenging ScanObjectNN (Uy et al., 2019a) contains $11,416$ training and $2,882$ validation point clouds of 15 categories, which are captured from the noisy real-world scenes and thus have domain gaps with the pre-trained ShapeNet (Chang et al., 2015) dataset. ScanObjectNN is divided into three splits for evaluation, OBJ-BG, OBJ-ONLY, and PB-T50-RS, where PB-T50-RS is the most difficult for recognition. ShapeNetPart (Yi et al., 2016) is a widely used dataset for semantic segmentation of 3D point clouds, which consists of

| Method | ModelNet40 |
|---|---|
| Transformer + OcCo (Yu et al., 2021) | 89.6 |
| Point-BERT (Yu et al., 2021) | 87.4 |
| Point-MAE (Pang et al., 2022) | 91.0 |
| Joint-MAE (Guo et al., 2023) | 92.4 |
| Point-M2AE (Zhang et al., 2022) | 92.9 |
| I2P-MAE (Zhang et al., 2023) | 93.4 |
| Ours + Point-MAE | 93.1 |
| **Ours + Point-M2AE** | **93.7** |

Table 10: Linear SVM Classification on ModelNet40 (Wu et al., 2015). We compare the accuracy (%) of existing self-supervised methods.

$16,881$ models across 16 categories, including objects such as chairs, tables, lamps, and airplanes. The ScanNet (Dai et al., 2017) is an indoor scene dataset consisting of $1,513$ reconstructed meshes, among which $1,201$ are training samples and 312 are validation samples.

## G  PSEUDO-CODE FOR MULTI-SCALE ATTENTION

Implementation details of Multi-Scale Attention mechanism are shown in the algorithm 1.

---
**Algorithm 1** Multi-Scale Attention Mechanism

---
1: **function** SCALEATTENTION($Q, K, V, \text{scale}_i$)
2:      $b, n, c \leftarrow \text{shape}(Q)$                                            ▷ Get dimensions
3:      $Q \leftarrow Q.\text{reshape}(-1, \text{scale}_i, c)$
4:      $K \leftarrow K.\text{reshape}(-1, \text{scale}_i, c)$
5:      $V \leftarrow V.\text{reshape}(-1, \text{scale}_i, c)$                              ▷ Partition inputs into scales
6:      $X \leftarrow \text{softmax}\left(\frac{Q \cdot K^{\mathsf{T}}}{\sqrt{c}}\right) \cdot V$                    ▷ Compute self-attention
7:      $X \leftarrow X.\text{reshape}(-1, n, c)$                            ▷ Reshape results
8:      **return** $X$
9: **end function**

---