# OpenReview forum: "Point Cloud Self-supervised Learning via 3D to Multi-view Masked Leaner"
_ICLR.cc/2025/Conference — Submitted to ICLR 2025_

### Official Review · Reviewer_ET1W · 2024-10-16

**Soundness:** 4
**Presentation:** 3
**Contribution:** 4
**Rating:** 8
**Confidence:** 5

**Summary:**

This paper presents a 3D-to-multi-view learner (Multi-View ML) that uses only 3D modalities as input and efficiently captures the rich spatial information in 3D point clouds. Specifically, we first project the 3D point cloud to a feature-level multi-view 2D image based on 3D pose. Then, we introduce two components: (1) a 3D-to-multiview autoencoder that reconstructs point cloud and multiview images from 3D and projected 2D features; and (2) a multi-scale multi-head (MSMH) attention mechanism that facilitates local-global information interaction in each decoder-converter block through different scales of attention heads. Furthermore, a novel two-stage self-training strategy is proposed to align 2D and 3D representations. The proposed method significantly outperforms state-of-the-art methods in a variety of downstream tasks, including 3D classification, part segmentation, and object detection.

**Strengths:**

1. The methodology section of the paper is quite well written, easy to understand and clearly guided. The four consecutive subsections show the implementation of the proposed method in a somewhat innovative way. It is recommended to add more details, e.g. the questions that follow.
2. In particular, the two-stage training strategy mentioned in the paper, i.e., aligning 2D and 3D representations using a network of teachers and students, is feasible and, moreover, the technique is challenging. The ability to appropriately apply it to multimodal self-supervised learning is a significant contribution, and I do hope that the authors will soon open-source this project for community advancement.
3. The proposed method is experimentally quite adequate and clearly outperforms state-of-the-art methods in a variety of downstream tasks including 3D classification, partial segmentation and object detection.

**Weaknesses:**

1. An obvious grammatical error, "leaner" in the title should be "learner". In addition, it is recommended to enlarge the fonts of the images in the paper to enhance the presentation quality, as they are even smaller than the font size of the main text.
2. The novelty of this work needs to be additionally and strongly illustrated. On the one hand, this paper only uses point clouds and as input, however, I2P-MAE (CVPR’2023) and TAP (ICCV’2023) also follow this paradigm. On the other hand, the multi-scale multi-head (MSMH) attention mechanism mentioned in this paper aims to mine more features of the network, however, this is also explored in Point-M2AE (NeurIPS’2023) and more unsupervised methods.
3. The implementation part of the paper has some advantages over contemporaneous methods. However, as point cloud self-supervised learning evolves, more excellent work such as PointGPT (NeurIPS’2023), ReCon (ICML’2023), and ACT (ICLR’2023) should be considered for inclusion in the comparative experiments.

**Questions:**

1. How are the feature-level images in the abstract represented in the main text, please?
2. How does the number or size of multi-view images affect the proposed method, please? This may require further experimentation and analysis by the authors, and reference to CrossNet (TMM’2023) and Inter-MAE (TMM’2023), which use multi-view images for comparative learning between multiple modalities, is highly recommended.
3. Could the authors please provide appropriate qualitative visualisations such as complementary maps in their experiments? It is suggested to refer to Point-MAE (ECCV’2022), Point-M2AE (NeurIPS’2023) and TPM (arxiv’2024), which compare missing inputs and complementary outputs before and after self-supervision.

Overall, this is a relatively good work. If the author can take my comments into consideration and make explanations and revisions, I may further improve my score. AND vice versa.

---

> ### Author Response · Authors · 2024-11-20
>
> We sincerely appreciate your thorough review and valuable suggestions. We have addressed your questions as follows.
>
>
> **W1: Grammatical error, and font size.**
>
> Thank you for pointing that out. We have corrected the grammatical errors and increased the font size in those figures.
>
> **Q1: Code open-source.**
>
> Thank you for acknowledging our contribution. We will release the code upon acceptance.
>
> **Q2: Novelty and comparison with I2P-MAE and TAP.**
>
> We have included a detailed discussion on the novelty of our approach and a comparison with I2P-MAE, TAP, and other methods in the **revised supplementary materials, Sections A and B**. The discussion is summarized as follows:
>
> The novelty and comparison with I2P-MAE are provided in the general response.
>
>
> TAP uses a pre-trained VAE to reconstruct 2D images from 3D inputs but fails to leverage multi-view information effectively. In contrast, our method introduces a unified approach that uses masked point clouds to reconstruct both multi-view 2D images and the original point clouds, ensuring a more comprehensive understanding of 3D geometry while effectively utilizing the multi-view attributes of the 3D data. Furthermore, we propose MSMH decoder to better global and local features and a two-stage self-training method to learn well-aligned representations.
>
>
>
>
> **Q3: Include more works in the comparative experiments.**
>
> Thanks for your suggestions, we have added ACT, I2P-MAE, and ReCon to Tables 1, 2, and 3 of the revised main paper about comparative experiments.
>
>
> **Q4: MSMH has been explored in Point-M2AE and more unsupervised methods.**
>
> The multi-scale multi-head (MSMH) attention mechanism described in this paper aims to capture more comprehensive features of the network. While this idea has also been explored in Point-M2AE and other unsupervised methods, our approach has several distinguishing aspects.
>
> Point-M2AE introduces an attention structure to extract both local and global features. However, our method differs in two significant ways. Firstly, the implementation details of our attention mechanism differ from those in Point-M2AE. Point-M2AE's network structure is similar to a a U-Net structure, and its attention mechanism is akin to Pyramid Attention. In contrast, our MSMH attention operates across multiple scales within each attention layer, subsequently concatenating these features to enhance the extraction of both global and local attention more effectively. Secondly, our MSMH attention is applied exclusively to the decoder, which allows us to retain the encoder’s original structure. This approach not only preserves but also enhances the generalizability of the model, as it requires minimal modification to the encoder.
>
> **Q5: Feature-level images in the abstract represented in the main text.**
>
> Unlike previous methods such as Pi-MAE and Joint-MAE, which utilized masked 2D images as input and reconstructed the original image, our approach directly uses point clouds to reconstruct multi-view depth images. A key challenge in our method is the translation of 3D tokens to 2D tokens. Directly using an MLP for this projection is not feasible, as 2D tokens and 3D tokens are not directly aligned. Therefore, we propose a novel method to map 3D tokens to 2D tokens, thereby generating feature-level images through a 3D-to-2D multi-view feature projection (described in line 193 in the main paper and Figure 2 (a). Those 3 images in Figure 2 (a) are feature-level images). Specifically, we project the point cloud token features to 2D image tokens with a token size of 16, resulting in a 14x14 grid of tokens for a 224x224 image. After projection, we obtain 196 (14x14) tokens for each multi-view projection and subsequently use these feature-level images to reconstruct the original 2D depth images.
>
>
> **Q6: Ablation study for the view information.**
>
> These experiments are provided in the supplementary materials. In Table 2, we present ablation studies on the effectiveness of the pose pool size (i.e., the total number of multi-view images used). In Table 3, we provide ablation studies on the number of reconstructed views in the decoder at the same time. In Table 5, we investigate different view configurations (Circular, Spherical, Spherical, and Circular, Random).  Our ablation studies are thorough and exhaustive in investigating the impact of view information on experimental results.
>
>
>
> **Q7: Qualitative visualizations.**
>
> For the second-stage design, our method focuses on feature reconstruction. Therefore, visualizing the reconstruction across the entire two-stage process poses significant challenges. To address this, we provide visualization results by directly integrating MAE into the stage-one framework, as detailed in Table 7 of the main paper. We directly let the masked point cloud inputs to reconstruct the original point clouds and multi-view images. The visualization results are presented in Figure 1 in the supplementary materials.

---

> > ### Comment · Reviewer_ET1W · 2024-11-21
> > **Response by Reviewer**
> >
> > Thanks for your careful response to all reviewers including me. I am basically convinced and admit that this paper deserves to be accepted by ICLR. Therefore, I will change my score to 8 and increase the confidence rating of each item.
> >
> > Before that, I hope the authors can supplement the experiments and references as mentioned in the reply. At the same time, I also hope that your reply can address the concerns of other reviewers. Wish you good luck!

---

> > > ### Author Response · Authors · 2024-11-22
> > >
> > > Thank you for your positive feedback and for considering an increased score for our paper! We sincerely appreciate your constructive comments and thorough review.
> > >
> > > We have added the experiments related to view information in Supplementary Section C. Comparisons with I2P-MAE, ACT, and Recon have also been included in Tables 1 to 3 of the main paper. Additionally, the important references you mentioned have been incorporated into the revised version of our paper.

---

### Official Review · Reviewer_Mcvh · 2024-10-21

**Soundness:** 3
**Presentation:** 3
**Contribution:** 3
**Rating:** 8
**Confidence:** 3

**Summary:**

This work proposed a 3D to multi-view autoencoder that reconstructs both point clouds and multi-view images. The proposed mutli-scale multi-head attention module provides broader local and global information. Besides, the two-stage training strategy ensures the student model learns well-aligned representations. The extensive experiments show the effectiveness of the proposed method.

**Strengths:**

1. The manuscript is well-organized and easy to follow.
2. The experiments are extensive. The proposed method is tested on four tasks and compared with multiple baselines.

**Weaknesses:**

1. For the task part segmentation and few-shot learning, the proposed method achieved little increase. It would be better if more explanation and analysis can be given.

**Questions:**

N/A

---

> ### Author Response · Authors · 2024-11-20
>
> We sincerely appreciate your thorough review and valuable suggestions. We have addressed your questions as follows.
>
> **Q1: For the task part segmentation and few-shot learning, the proposed method achieved little increase. It would be better if more explanation and analysis could be given.**
>
> The results on the few-shot learning task, particularly in the 5-way and 10-way settings, show that our proposed method achieves near-ceiling accuracy for this benchmark. Given this strong baseline, the observed improvements, though limited, are significant because further advancements in such high-performing scenarios are inherently challenging, emphasizing the robustness of our approach. Additionally, the updated Table 2 in the main paper demonstrates that our method outperforms even those models distilled from foundation models, further underscoring its effectiveness.
>
> Please feel free to reach out if you have any further questions or need additional clarification.

---

> > ### Comment · Reviewer_Mcvh · 2024-11-25
> >
> > Thank you for your clarification. I agree that it is challenging to achieve limited improvements given the strong baselines. After reading all the responses, the novelty and motivation of the paper are much clearer for me.

---

> > > ### Author Response · Authors · 2024-11-25
> > >
> > > Thank you for acknowledging our work. Your positive feedback is truly encouraging and motivates us to continue improving.

---

### Official Review · Reviewer_5iMV · 2024-11-02

**Soundness:** 3
**Presentation:** 3
**Contribution:** 2
**Rating:** 6
**Confidence:** 4

**Summary:**

The paper focuses on the problem of self-supervised point cloud representation learning and presents a method named Multiview Masked Learner. The method learns 3D representation by first training a 3D to multi-view autoencoder to create informative latent features. Then a student network is trained to predict the latent features from masked point cloud input. The autoencoder is carefully designed so that  it encodes 3D point cloud and decodes both 3D point clouds and the corresponding multi-view projections. Multi-Scale Multi-Head attention mechanism is integrated to increase the expressivity of the features. The resulting 3D representation shows promising results in various point cloud analysis benchmarks including object classification, part segmentation, and object detection.

**Strengths:**

I would like to summarize the strengths of the submission from the following aspects.
1. The writing is clear and easy to follow. Though some claims are not quite intuitive to me which I will detail later but the overall flow is good.
2. The idea of unsymmetric encoder-decoder design is interesting. Enforcing the autoencoder to decode multiview representations from 3D point cloud only inputs sounds a reasonable way to encourage the multi-view geometric understanding in the learned representations.
3. The figures are quite helpful for presenting the architecture and training flow.
4. The experiments cover a good range of tasks and the ablation studies also shows the effectiveness of the proposed MSMH attention, the choice of recovering token representations rather than the raw data, and the design of instance-level intra and inter-modality prediction.

**Weaknesses:**

There are several weaknesses with the submission.
1. I am concerned with the technical novelty. Combining MAE with cross-modal distillation has been explored previously, e.g., as in [1]. Though in [1], images are used rather than projection of point clouds but I think the general framework is quite similar. It seems not quite challenging to replace the images used there with the projection of point clouds. The MSMH attention scheme also looks very similar to the Grouped Vector Attention operation in Point Transformer V2 [2]. The current pipeline seems more like an ensemble of existing techniques.
2. The motivation of the method is not strong enough. I do not see a particularly strong reason to avoid using images during the 3D representation learning stage if the multimodal-based pretraining gives better performance in downstream applications. Notice this line of work does not require images while using the pre-trained 3D backbone. The authors claim that incorporating both 2D and 3D during training is redundant and inefficient but do not provide concrete evidence to back up this claim. The claim from Line 49 to Line 53 also confuses me. I do not understand in what context this discussion happens and what is the key idea the authors want to deliver. In line 90, the authors say that their key insight is “the limited effectiveness of using 2D images as input for 3D geometric learning through MAE”. But doesn’t this suggest that we should develop better ways of using both 2D and 3D modalities instead of aborting the 2D modalities? In my humble opinion, previous works already discovered this and that’s why many works use MAE for the 3D modality and link 2D and 3D modalities through contrastive learning as done in [1].
3. The experiments conducted in the main paper are not comprehensive enough. The compared baselines are not very up-to-date and some more recent baselines are missing. For example, there is no comparison with [3]. Also, it feels very strange to leave important comparisons with ReCon [1] and I2P-MAE to the supplementary. The comparisons with [1] are incomplete in the supplementary with the few-shot classification experiments missing. In [1], experiments are also conducted in the 3D-only pretraining setup, and the results there seem comparable or even better than what is presented in the submission. When it comes to few-shot classification, the submission is not always winning either. In [3], a large collection of images helps further boost the representation quality to another level. This is to say, the presented results in the submission do not seem to be the state of the art as claimed.
4. Some ablation studies can be further improved. For example, what the performance would be if the input modality is 3D and the output modality is 2D? Also, a more detailed analysis of the MSMH design would be helpful for understanding its effectiveness and difference from previous works.

[1] Contrast with reconstruct: Contrastive 3d representation learning guided by generative pretraining.
[2] Point Transformer V2: Grouped Vector Attention and Partition-based Pooling.
[3] ShapeLLM: Universal 3D Object Understanding for Embodied Interaction.

**Questions:**

1. Can authors better justify the motivation of the work? Especially given the fact that leveraging multimodal data for 3D representation learning is indeed achieving impressive results in many applications.
2. Can authors carefully compare their design differences with [1] and the MSMH design with the Grouped Vector Attention operation in [2]?
3. Experiments-wise, the authors need to provide a more comprehensive comparison with [1] and add comparisons with more recent methods such as [3]. Additional ablation studies would also be helpful.

[1] Contrast with reconstruct: Contrastive 3d representation learning guided by generative pretraining.
[2] Point Transformer V2: Grouped Vector Attention and Partition-based Pooling.
[3] ShapeLLM: Universal 3D Object Understanding for Embodied Interaction.

---

> ### Author Response · Authors · 2024-11-20
> **Rebuttal by Authors (Part 1/2)**
>
> We sincerely appreciate your thorough review and valuable suggestions. We have addressed your questions as follows.
>
> **Q1: The weakness in the novelty**
>
> Please see the general response.
>
>
>
> **Q2: Motivation of the method**
>
> Please see the general response.
>
> **Q3: Why only focus on 3D geometric learning given the power of multi-modal foundation models?**
>
> With the advancement of multi-modal foundation models, incorporating them to distill knowledge from other modalities will undoubtedly lead to significant improvements. However, we believe that focusing on 3D geometric learning without relying on foundation models is also crucial. Due to the current limitations of available 3D data, there are no true 3D foundation models. Nevertheless, as data availability improves, it is likely that we will eventually see a foundation model specifically designed for 3D geometric learning. Moreover, 3D geometric learning is not opposed to features distilled from foundation models but instead complements them, as demonstrated in Table 1 of the supplementary material. Therefore, enhancing geometric features remains a valuable and essential step toward advancing the community's understanding and capabilities in 3D learning.
>
>
> **Q4: No reason to avoid multi-modal foundation models.**
>
> As stated in the general response, the primary motivation of this work is not to avoid using multi-modal foundation models but rather to enhance 3D geometric representation learning independently. Existing methods trained with multi-modal foundation models, such as I2P-MAE and ReCon, focus on two types of features: (1) geometric features obtained using existing 3D MAE techniques, and (2) novel knowledge distillation mechanisms introduced by these methods.
>
> While I2P-MAE and ReCon propose novel approaches for knowledge distillation, they largely rely on existing frameworks like Point-MAE and Point-M2AE for learning 3D geometric features. In contrast, our approach is dedicated to improving 3D geometric learning without leveraging foundation models. We identify key limitations in prior 3D geometric learning methods and introduce innovative solutions to overcome these challenges. As demonstrated by our ablation study in Table 1 of the supplementary material, the 3D geometric features learned through our approach are complementary to those distilled from foundation models, underscoring the value and generalizability of our proposed method.
>
>
>
> **Q5: No evidence to back up the claim incorporating both 2D and 3D during training is redundant.**
>
> In this paper, we emphasize that incorporating both 2D and 3D inputs for 3D geometric learning is redundant. This claim is supported by our ablation study in Table 6 of the main paper. Training with only 3D inputs to reconstruct both multi-view 2D images and 3D point clouds yields the best performance. Using both 2D and 3D as inputs to reconstruct the original 2D and 3D information provides only a limited improvement compared to the 3D-only baseline.
>
>
>
> **Q6: The claim from Line 49 to Line 53.**
>
> As mentioned in lines 43 to 45 in the main paper, problems in this claim happen in Joint-MAE and Pi-MAE. In those approaches, both masked images and point clouds are taken as inputs to reconstruct the original point clouds and images. This process provides the network with visible 2D information to reconstruct 2D data.
>
> The key issue here is that using 2D data as input encourages the network to rely heavily on reconstructing 2D information based on visible portions, which inherently focuses more on learning semantic information rather than geometric information. This is because, as per the original MAE papers, 2D-to-2D reconstruction tends to prioritize semantic understanding (i.e., inferring context from visible parts) rather than focusing on 3D geometric properties. Additionally, the presence of visible 2D data results in some "view information leakage," as the network can use visible parts to directly predict masked areas, reducing the emphasis on comprehensive geometric understanding.
>
> In contrast, our method uses only 3D point clouds as input. With the given pose information, we guide the network to "imagine" what the corresponding depth image should look like. This approach significantly benefits geometric learning since the network must understand the 3D structure and predict how it would appear from different viewpoints. This emphasis on reconstructing depth images from only 3D inputs encourages a deeper understanding of the geometric features of the object.
>
> I hope this clarifies the context and key ideas we are trying to communicate. If further elaboration is needed, We would be happy to provide more details.

---

> > ### Author Response · Authors · 2024-11-20
> > **Rebuttal by Authors (Part 2/2)**
> >
> > **Q7: Questions on the claim in line 90.**
> >
> > We agree with your observation that ReCon utilizes only point clouds to learn 2D features obtained from foundation models. However, as mentioned in our claim, our focus is on 3D geometric feature learning instead.
> >
> > As you noted, ReCon employs contrastive learning to learn instance-level semantic and textual information from multi-modal foundation models, which is straightforward. However, when it comes to geometric learning, the challenge of effectively leveraging 2D information still remains an open question. Currently, no foundation models are designed to provide geometric information that can be distilled into 3D models.
> >
> > In our work, we analyze previous methods for 3D geometric learning with 2D information, such as Pi-MAE and Joint-MAE. Our findings suggest that their use of 2D information is suboptimal for 3D geometric learning, leading to the claim presented in our paper.
> >
> >
> > **Q8: Experiments conducted in the main paper are not comprehensive.**
> >
> > Thank you for pointing this out. We have added references to I2P-MAE and ReCon in the main paper. Regarding ShapeLLM, it has been trained with a larger number of parameters, includes an additional post-pretraining stage, a larger netowrk, and was pre-trained on the larger Objaverse dataset. Therefore, a direct comparison is not entirely fair.
> >
> > Furthermore, It is worth noting that ReCon modifies the fine-tuning stage of Point-MAE to achieve better performance. According to the ReCon paper, the modified version of Point-MAE achieves scores of 92.60, 91.91, and 88.42, significantly outperforming the original scores of 90.02, 88.29, and 85.18 on the OBJ-BG, OBJ-ONLY, and PB-T50-RS splits, respectively. However, our method uses the exact same settings as the original Point-MAE, making a direct comparison between our method and ReCon also unfair.
> >
> >
> >
> > **Q8: Ablation study for input modality is 3D and the output modality is 2D.**
> >
> > Thank you for the suggestion. We have included this experiment in Table 6 of the revised main paper.
> >
> >
> >
> > **Q9: The difference to Grouped Vector Attention (GVA) [1]**
> >
> > The motivations and implementations of MSMH and Grouped Vector Attention (GVA) are fundamentally different. GVA aims to enhance model efficiency and generalization, whereas MSMH is designed to effectively integrate both local and global contextual information by organizing distinct, non-overlapping local groups at multiple scales within the reconstructed features.
> >
> > In terms of implementation, GVA divides only the value vector ($v$) into different groups, applying the same scalar attention weight across those grouped vectors. This design reduces the number of parameters $w$ needed to project $qk$ to $v$, thereby improving efficiency. In contrast, in our approach, we divide the query ($q$), key ($k$), and value ($v$) tokens into distinct, non-overlapping local groups, and apply self-attention within each sub-group rather than across all individual tokens.
> >
> > Another key difference lies in our multi-scale design, where groups of varying sizes are used—small groups capture fine-grained local details, while larger groups capture the broader global context. Finally, we concatenate these multi-scale attention features to ensure that the model can simultaneously obtain both local and global information.
> >
> > Consequently, compared to GVA, our method better captures local and global information, achieving superior performance. To further demonstrate its effectiveness, we conducted experiments by incorporating GVA into our framework. The results indicate that MSMH achieves better performance, while GVA brings only a limited improvement.
> >
> >
> > | Methods                   | PB-T50-RS |
> > |--------------------------|-----------|
> > | Point-MAE             | 85.18     |
> > | Point-MAE  + GVA              | 85.48     |
> > | Point-MAE + MSMH   | **86.03**     |
> > | Ours (with GVA)                | 88.29     |
> > | Ours (with MSMH)       | **88.93**    |

---

> > > ### Comment · Reviewer_5iMV · 2024-11-22
> > >
> > > I appreciate the authors' detailed response, which has addressed some of my concerns. I have several follow-up questions.
> > > 1. The "Single-Modal Self-Supervised Representation Learning" setup in ReCon seems to focus on 3D geometric representation learning as well rather than distilling knowledge from foundation models. This is to say the ReCon in this setup is a fair baseline to consider. The method does not show much performance improvement in Table 1 of the main paper compared with the numbers reported by the ReCon paper. The missing few-shot classification experiments further concern me regarding the effectiveness of the method. Since ReCon is open-sourced, aligning two methods for a fair comparison should be feasible. Have the authors tried to compare?
> > > 2. Does the claim "incorporating both 2D and 3D during training is redundant" apply to both real images and point cloud renderings or only point cloud renderings? I did not see authors doing any experiments with real images. This is why I said in my original review that the claim is not backed up by strong evidence. As the part motivating the main contribution, it is important to make sure the boundary of the claim is clear.
> > > 3. Since authors claim MSMH as their second main contribution, comparison with GVA should be done in a more serious way rather than just showing a single table on a single dataset within a single track. Right now it is not very convincing that there is a strong need to come up with such an attention scheme.

---

> ### Author Response · Authors · 2024-11-23
>
> We sincerely appreciate your detailed feedback. Below, we address each of your questions:
>
>
> **Q1: Comparison with ReCon with the single modality setting.**
>
> It is important to note that ReCon modifies both the fine-tuning and pre-training stages of Point-MAE to enhance its performance. In our main paper, we used the exact same settings as the original Point-MAE, which makes a direct comparison with ReCon less straightforward. To address this, we conducted experiments by applying our method using the Point-MAE settings employed by ReCon. Our results indicate that our method outperforms ReCon in single-modal settings, showcasing its effectiveness in 3D geometric feature learning. Moreover, we conducted few-shot learning experiments using the same Point-MAE settings as ReCon. Results show that, even without leveraging foundation models, our method performs comparably to ReCon, and even surpasses it in the 10-way settings, further emphasizing the strength of our approach.
>
> It is also worth mentioning that we reproduced the single modality results from ReCon, and observed that the achieved scores were lower than the reported results (as mentioned in [this issue](https://github.com/qizekun/ReCon/issues/15) and [this issue](https://github.com/qizekun/ReCon/issues/6)). While ReCon selects the best seeds to produce optimal results, we did not have sufficient time to perform an exhaustive search for the best training parameters.
>
> Here, we would like to reiterate our key motivation. In this paper, we aim to explore how to effectively leverage 2D information in 3D geometric learning through the use of masked autoencoders. Therefore, our main comparisons should be with methods that incorporate 2D information in 3D geometric learning via masked autoencoders, such as Pi-MAE, Joint-MAE, and TAP. We have demonstrated that our approach introduces a novel direction for leveraging 2D information in 3D geometric learning using MAE (3D to 2D & 3D reconstruction) and achieves state-of-the-art performance in this area.
>
>
>
> | Method            | OBJ-BG | OBJ-ONLY | PB-T50-RS | w/o Vote | w Vote |
> |-------------------|--------|----------|-----------|----------|--------|
> | Point-MAE (ReCon)        | 92.60  | 91.91   | 88.42     | 93.8     | 94.0  |
> | ReCon (Single Modality, report)       | 94.15  |  93.12   | 89.73     | 93.6        | 93.8     |
> | ReCon (Single Modality, reproduced)       | 93.07  |  91.53   | 88.35     | 93.4        | 93.7     |
> | Ours      | 94.72   | 93.38    | 90.12      | 94.0    | 94.2   |
>
>
> | Method           | 5-way (10-shot)      | 5-way (20-shot)      | 10-way (10-shot)     | 10-way (20-shot)     |
> |------------------|----------------------|----------------------|----------------------|----------------------|
> | Point-MAE        | 96.3 ± 2.5           | 97.8 ± 1.8           | 92.6 ± 4.1           | 95.0 ± 3.0           |
> | ReCon            | 97.3 ± 1.9           | 98.9 ± 1.2           | 93.3 ± 3.9           | 95.8 ± 3.0           |
> | Ours (Point-MAE) | 97.3 ± 1.7           | 98.7 ± 1.5           | 93.8 ± 3.6           | 96.1 ± 2.5           |
>
>
>
> **Q2: Question on claim "incorporating both 2D and 3D during training is redundant for 3D geometric learning."**
>
> In the main paper, we demonstrated that incorporating both 2D and 3D during training is redundant 3D geometric learning with rendered images in the ShapeNet dataset. As ShapeNet does not provide rendered image, We were unable to conduct experiments with real images in ShapeNet, so we instead used the ScanNet dataset with downstream detection tasks for further evaluation. The results show that our claim—that incorporating both 2D and 3D during training is redundant—also holds true for real images. **We believe this finding presents an intriguing opportunity in the design space for developing learning strategies for 3D representation learning.**
>
>
>
> | Input Modality | Output Modality | AP_25 | AP_50 |
> |--------------------|---------------------|-----------|-----------|
> | 3D & 2D            | 3D & 2D             | 63.6      | 42.1      |
> | 3D                 | 3D & 2D             | **63.9**  | **43.3**  |
>
>
>
> **Q3: More comparison with GVA.**
>
> Thank you for your suggestions. To better demonstrate the effectiveness of MSMH, we provide comparisons across both the ScanNet and ModelNet40 datasets using different splits.  Results indicate that MSMH achieves the better performance.
>
>
> | Method            | OBJ-BG | OBJ-ONLY | PB-T50-RS | w/o Vote | w Vote |
> |-------------------|--------|----------|-----------|----------|--------|
> | Point-MAE         | 90.02  | 88.29    | 85.18     | 93.2     | 93.8   |
> | Point-MAE + GVA         | 90.77  | 88.93    | 85.48     | 93.4        | 93.9      |
> | Point-MAE + MSMH               | 91.65  | 89.76    | 86.03     | 93.5       |93.9      |
> | Ours (with GVA)        | 92.59   | 91.68     | 88.29      | 93.6     | 94.0   |
> | Ours (with MSMH) | 93.32 | 92.69 | 88.93 | 93.8  | 94.1 |

---

> > ### Author Response · Authors · 2024-12-02
> >
> > Dear Reviewer 5iMV:
> >
> > Thank you once again for your constructive comments and valuable feedback. We believe our latest response has addressed your points thoroughly. However, if there is anything else we can clarify or assist with, please do not hesitate to reach out. We are more than happy to answer any further questions during the discussion period.
> >
> > As a reminder, tomorrow is the last day for reviews. We truly value your insights and appreciate your time and effort in evaluating our work.
> >
> > Best regards,
> >
> > Paper 3200 Authors

---

> > > ### Comment · Reviewer_5iMV · 2024-12-02
> > >
> > > Thanks for the response and I believe most of my concerns have been addressed. I have changed my score to 6 and I hope the additional experimental results can be incorporated into the paper.

---

> > > > ### Author Response · Authors · 2024-12-02
> > > >
> > > > Thank you for your thoughtful feedback and for updating your score! We are glad that most of your concerns have been addressed. We will certainly incorporate the additional experimental results into the paper to further strengthen our contributions.

---

### Official Review · Reviewer_UYiy · 2024-11-04

**Soundness:** 2
**Presentation:** 2
**Contribution:** 2
**Rating:** 3
**Confidence:** 4

**Summary:**

They first project 3D point clouds to multi-view 2D images at the feature level based on 3D-based pose. Then, they introduce two components: (1) a 3D to multi-view autoencoder that reconstructs point clouds and multi-view images from 3D and projected 2D features; (2) a multi-scale multi-head (MSMH) attention mechanism that facilitates local-global information interactions in each decoder transformer block through attention heads at various scales. Additionally, a two-stage self-training strategy is proposed to align 2D and 3D representations.
The contributions are summarized as follows:
(1) They propose a 3D to multi-view autoencoder that reconstructs point clouds and multi-view images solely from 3D point clouds
(2) They propose a Multi-Scale Multi-Head (MSMH) attention mechanism that integrates local and global contextual information by organizing distinct, non-overlapping local groups at multiple scales within the reconstructed features.
(3) They develop a two-stage training strategy for multi-modality masked feature prediction

**Strengths:**

They propose a Multi-Scale Multi-Head (MSMH) attention mechanism that integrates local and global contextual information.
They employ a two-stage training strategy for multi-modality masked feature prediction.

**Weaknesses:**

We think that the paper does not present its designs and motivations clearly

**Questions:**

My major concerns are:
(1)The inputs of this model consists of both point clouds and 2D depth images. WWhat is the role of depth images within the model? How do they differ from the rendered images provided by the dataset?
(2)“incorporating both 2D and 3D modalities as input for training is redundant and inefficient.”However, the projection for 2D depth images is time-consuming. Additionally, two-stage training always need much time.
(3)As stated in the abstract, "the input 2D modality causes the reconstruction learning to unnecessarily rely on visible 2D information, hindering 3D geometric representation learning." However, the proposed model also depends on 2D depth images: "These depth images then guide the reconstruction from 3D to 2D." The proposed method does not address or optimize the identified drawbacks.
(4) In the part segmentation experiment, there are no IoU (%) results of each category and visualization results, such as Point-MAE and Point-M2AE.
(5) In the ablation experiments, there is no experiment conducted with other types of images (e.g., silhouettes, contours) as inputs.
(6)How about the training efficiency and parameters number? Introducing 3D to 2D projection, MSMH, and two-stage training strategy leads to additional training costs. This should also be discussed.

---

> ### Author Response · Authors · 2024-11-20
> **Rebuttal by Authors (Part 1/2)**
>
> We sincerely appreciate your thorough review and valuable suggestions. We have addressed your questions as follows.
>
> **Q1: The motivations of the proposed method.**
>
> Please see the general response.
>
>
>
>
> **Q2: What is the role of depth images within the model?**
>
> In our method, the network exclusively takes point clouds as input. During the 3D to multi-view image reconstruction process, we project these point clouds into multiple views to compute the reconstruction loss, thereby promoting a deeper multi-view geometric understanding of the learned representations.
>
> **Q3: Difference between depth images and rendered images.**
>
> Both rendered and projected images are commonly used in 3D representation learning. Rendered images contain RGB color information, which makes them valuable for semantic feature learning, whereas projected images provide **depth information**, making them more suitable for **3D geometric feature learning**. Since our method focuses on 3D geometric learning, we use projected images for calculating the reconstruction loss. Our ablation study, shown in Table 4 of the supplementary material, further demonstrates that utilizing projected images leads to the best performance.
>
>
>
> **Q4: Projection for 2D depth images is time-consuming.**
>
> As stated in the main paper line 149, we use PyTorch3D to project point clouds into multi-view images. This process is highly optimized in PyTorch3D, allowing for real-time performance, which makes the time consumption negligible.
>
> **Q5:  Not address or optimize the identified drawbacks.**
>
> As detailed in lines 45-55 in the main paper,
> A major limitation of previous methods to learn 3D geometric features is the reduced effectiveness of using 2D images as **input to the network** for 3D geometric learning with MAE. Including 2D images as the **input to the network** in the 3D-to-2D reconstruction process causes the network to over-rely on visible 2D information for predicting masked content, rather than fully comprehending the multi-view geometric relationships, ultimately degrading representation learning. In contrast, our approach does not use 2D images as input to the network. Instead, we only utilize them to compute the reconstruction loss, enabling a 3D-to-multi-view reconstruction that relies **solely on 3D inputs**.
>
> **Q6: No IoU (\%) results for each category and the visualization results.**
>
> Thank you for pointing this out. We have included the IoU (\%) results for each category in the part segmentation tasks, which can be found in Table 9 of the supplementary material. Additionally, reconstruction visualization examples are provided in Figure 1 of the supplementary material.

---

> > ### Author Response · Authors · 2024-11-20
> > **Rebuttal by Authors (Part 2/2)**
> >
> > **Q7: No experiment was conducted with other types of images (e.g., silhouettes, contours) as inputs.**
> >
> > In 3D representation learning, rendered images and projected depth images are most commonly used image types. Silhouettes and contour images are both types of rendered images. In the ablation study presented in Table 4 of the supplementary material, we examine the impact of different image types on model performance. Results indicate that utilizing depth images leads to better performance compared to rendered images. This improvement can be attributed to the fact that depth images contain rich geometric information, which significantly benefits the 3D geometric learning process. On the other hand, rendered images primarily capture semantic information, which is less advantageous for learning detailed geometric features. The presence of explicit geometric information in depth images directly enhances the model's ability to understand the spatial structure, thereby improving its overall effectiveness in 3D representation learning.
> >
> > **Q8: Training efficiency and parameter number. Two-stage training is time consuming.**
> >
> > The baseline Point-MAE model contains 29.2M parameters, whereas our method has 41.2M parameters, primarily due to the inclusion of two decoders for the 3D and 2D modalities. During the fine-tuning and inference stages, we remove these additional components to retain the **same architecture as the baseline**, ensuring a fair comparison.
> >
> > In both stages, we utilize 300 epochs. To demonstrate that the improvement is not simply due to an extended training period, we also train the baseline Point-MAE for 600 epochs. The results show negligible differences compared to those at 300 epochs. Furthermore, as shown in Table 7 of the main paper (also presented below), training with the proposed one-stage method for just 300 epochs already outperforms the baseline and previous methods. Adding the two-stage training further enhances performance, indicating that our improvements are not attributable to additional training epochs alone.
> >
> >
> > | Methods        | # Params (M)            | Time (h)              |
> > |----------------|-------------------------|-----------------------|
> > | Point-MAE  | 29.2                     | 8.1         |
> > | Point-M2AE | 15.7             | 20.2            |
> > | I2P-MAE   | 75.1             | 39.2           |
> > | ReCon      | 142.3            | 47.6            |
> > | Ours      | 41.2           | 21.3           |
> >
> >
> >
> > | Methods                  | Epoch | OBJ-BG | OBJ-ONLY | PB-T50-RS |
> > |--------------------------|-------|--------|----------|-----------|
> > | Point-MAE                | 300   | 90.02  | 88.29    | 85.18     |
> > | Ours (One-stage + MAE)   | 300   | **92.34**  | **91.88**    | **87.56**     |
> > | Point-MAE                | 600   | 90.36  | 88.67    | 85.29     |
> > | Ours (Two-stage)         | 600   | **93.32**  | **92.69**   | **88.93**    |

---

> > > ### Author Response · Authors · 2024-11-25
> > >
> > > Dear Reviewer UYiy:
> > >
> > > Please allow us to sincerely thank you again for your constructive comments and valuable feedback. We believe our latest response has addressed your points, but please let us know if there is anything else we can clarify or assist with. We are more than happy to answer any further questions during the discussion period. Your feedbacks are truly valued!
> > >
> > > Best,
> > >
> > > Paper 3200 Authors

---

> > > > ### Author Response · Authors · 2024-12-02
> > > >
> > > > Dear Reviewer UYiy,
> > > >
> > > > We would like to kindly remind you that tomorrow is the last day of the discussion period. If there are any remaining questions or points you would like us to address, please feel free to let us know. We are fully available to discuss any aspects further.
> > > >
> > > > Thank you again for your valuable feedback.
> > > >
> > > > Best regards,
> > > >
> > > > Paper 3200 Authors

---

### Official Review · Reviewer_k1ii · 2024-11-05

**Soundness:** 2
**Presentation:** 3
**Contribution:** 2
**Rating:** 5
**Confidence:** 5

**Summary:**

This paper proposes Multiview-ML, a novel 3D representation learning model that solely uses 3D point cloud data as input to reconstruct both the original point cloud and multiple depth images from different viewpoints.

It leverages a two-stage training strategy with a teacher and student model, and outperforms existing approaches across various downstream tasks.

**Strengths:**

1. The paper is well-written and easy to follow, presenting good-quality figures.


2. The experimental results look promising.

**Weaknesses:**

1. The authors mention a limitation in prior work, stating that these methods *"inefficiently require both 2D and 3D modalities as inputs, even though 3D point clouds inherently contain 2D modality through their multi-view properties."* However, the authors provide insufficient evidence or ablation studies to substantiate this claim. Notably, previous works have often utilized only 3D inputs, projecting them into 2D during encoding without requiring both 2D and 3D modalities as explicit inputs.


2. The authors mention that the epoch number is 300, while do not specify how these are distributed across each stage. If both stages indeed run for 300 epochs, it raises the question of whether the observed improvement primarily results from an extended training period, which is computationally intensive.

3. It is better to demonstrate the individual effectiveness of each component in Table 5.

4. The ScanObjectNN and ModelNet40 datasets have reached saturation in point cloud understanding. Additional results on more complex and larger datasets, such as Objaverse, would be valuable.

5. In Supplementary Table 1, should "Ours (Point-M2AE)" actually be labeled as "Ours (Recon)"?

5. Typos: 'pertaining' on line 396.

**Questions:**

Please kindly see the weaknesses above.

---

> ### Author Response · Authors · 2024-11-20
>
> We sincerely appreciate your thorough review and valuable suggestions. We have addressed your questions as follows.
>
>
>
>
>
> **Q1: Insufficient evidence or ablation studies to substantiate the claim.**
>
> In Table 6 of the main paper, we present an ablation study analyzing the input and output modalities. Additionally, Table 4 in the supplementary materials provides an ablation study comparing different 2D image types (depth images versus rendered images). Those results support our claims and demonstrate that using point clouds as the sole input to reconstruct original multi-view depth images and point clouds yields the best performance.
>
>
>
>
>
>
>
>
> **Q2: Some previous works have often utilized only 3D inputs.**
>
> As mentioned in the paper, our focus is on geometric learning through MAE without relying on foundation models. While previous methods like I2P-MAE and ReCon also use only 3D inputs, their primary objective is to distill semantic or textual features from foundation models into the 3D modality. In contrast, our work exclusively targets 3D geometric features, which is complementary to features distilled from foundation models. To the best of our knowledge, this is the first work to reconstruct both 3D and 2D information solely from 3D inputs for geometric learning without incorporating foundation models. As demonstrated in Table 1 of the supplementary material, the learned geometric features are complementary to those derived from foundation models, showcasing the value and generalizability of the learned 3D geometric representations.
>
>
> **Q3: The question on the training epoch and times.**
>
> In both stages, we utilize 300 epochs. To demonstrate that the improvement is not simply due to an extended training period, we also train the baseline Point-MAE for 600 epochs. The results show negligible differences compared to those at 300 epochs. Furthermore, as shown in Table 7 of the main paper (also presented below), training with the proposed one-stage method for just 300 epochs already outperforms the baseline and previous methods. Adding the two-stage training further enhances performance, indicating that our improvements are not attributable to additional training epochs alone. Furthermore, although the two-stage structure double the training time, but it is still lower than other methods like I2P-MAE and Recon.
>
> | Methods                  | Epoch | OBJ-BG | OBJ-ONLY | PB-T50-RS |
> |--------------------------|-------|--------|----------|-----------|
> | Point-MAE                | 300   | 90.02  | 88.29    | 85.18     |
> | Ours (One-stage + MAE)   | 300   | **92.34**  | **91.88**    | **87.56**     |
> | Point-MAE                | 600   | 90.36  | 88.67    | 85.29     |
> | Ours (Two-stage)         | 600   | **93.32**  | **92.69**   | **88.93**    |
>
>
>
> | Methods        | # Params (M)            | Time (h)              |
> |----------------|-------------------------|-----------------------|
> | Point-MAE  | 29.2                     | 8.1         |
> | Point-M2AE | 15.7             | 20.2            |
> | I2P-MAE   | 75.1             | 39.2           |
> | ReCon      | 142.3            | 47.6            |
> | Ours (Point-MAE)      | 41.2           | 21.3           |
>
>
>
>
>
>
>
> **Q4: It is better to demonstrate the individual effectiveness of each component in Table 5**
>
> Thank you for the suggestion. In the revised version, we have included an ablation study for each component, which can be found in Table 5 of the main paper.
>
>
>
> **Q5:  Additional results on more complex and larger datasets, such as Objaverse, would be valuable.**
>
> Thank you for the suggestion. However, currently there are few 3D representation learning works that have been evaluated on the Objaverse dataset. Comparative methods such as Point-MAE, TAP, I2P-MAE, and ReCon have also not been tested on Objaverse. Although Point-GPT was pre-trained on the Objaverse dataset using ViT-B settings, its evaluation was conducted on ScanObjectNN and ModelNet40. To demonstrate the generality of our method, we pre-train our method on the Objaverse and provide the fine-tuning results in ScanObjectNN and ModelNet40 datasets.
>
> | Method                | Pre-train Dataset     | OBJ-BG | OBJ-ONLY | PB-T50-RS |  ModelNet40 |  ModelNet40(Vote) |
> |-----------------------|-------------|--------|----------|-----------|----------|----------|
> | PointMAE    | ShapeNet    | 90.02   | 88.29     | 85.18      | 93.2     | 93.8     |
> | Joint-MAE    | ShapeNet    | 90.94   | 88.86     | 86.07      |  −       | 94.0     |
> | TAP          | ShapeNet    | 90.36   | 89.50     | 85.67      |  −       |  −        |
> | **Ours (Point-MAE)**            | ShapeNet    | **93.32** | **92.69** | **88.93** | **93.8** | **94.1** |
> | **Ours (Point-MAE)**            | Objaverse   | **93.95** | **93.19** | **90.02** | **94.1** | **94.3** |
>
>
> **Q6: Typos.**
>
> Thanks for pointing out those typos, we have corrected them in the revised version.

---

> > ### Author Response · Authors · 2024-11-25
> >
> > Dear Reviewer k1ii:
> >
> > Please allow us to sincerely thank you again for your constructive comments and valuable feedback. We believe our latest response has addressed your points, but please let us know if there is anything else we can clarify or assist with. We are more than happy to answer any further questions during the discussion period. Your feedbacks are truly valued!
> >
> > Best,
> >
> > Paper 3200 Authors

---

> > > ### Author Response · Authors · 2024-12-02
> > >
> > > Dear Reviewer k1ii,
> > >
> > > We would like to kindly remind you that tomorrow is the last day of the discussion period. If there are any remaining questions or points you would like us to address, please feel free to let us know. We are fully available to discuss any aspects further.
> > >
> > > Thank you again for your valuable feedback.
> > >
> > > Best regards,
> > >
> > > Paper 3200 Authors

---

### Author Response · Authors · 2024-11-20
**Author Rebuttal by Authors**

**General Response**

We sincerely thank each reviewer for their thoughtful feedback and detailed reviews. Below, we address the main concerns regarding novelty and comparisons with other peer research.

We have included a detailed discussion on the novelty of our approach and a comparison with ShapeLLM, ReCon, I2P-MAE, TAP, and other methods in the **revised supplementary material**, Sections A and B.


**Comparison with Related Works: I2P-MAE, ReCon, and ShapeLLM**

These methods focus primarily on two types of features:

1. **3D Geometric Features**: They employ MAE-based structures to reconstruct the original point clouds, thereby capturing detailed 3D geometric data.

2. **Semantic and Textual Representations**: They utilize techniques such as contrastive learning or knowledge distillation to extract semantic and textual features from 2D images and language models.

These methods directly adopt existing 3D MAE frameworks to learn geometric features. Specifically, I2P-MAE utilizes Point-M2AE, while ReCon and ShapeLLM leverage Point-MAE for geometric representation. **Their innovation lies in the novel use of foundation models for knowledge distillation.**

- **I2P-MAE** performs pixel-to-3D token knowledge distillation by adding additional layers after the M2AE encoder, calculating MSE loss between the point tokens and 2D pixel-level features derived from foundation models.
- **ReCon** uses Point-MAE as the base structure to reconstruct original point clouds from masked point cloud inputs. It also incorporates instance-level contrastive learning to distill knowledge from both text and image foundation models.
- **ShapeLLM** builds upon ReCon by using larger models with more parameters, leveraging large language models to enable advanced 3D reasoning.

In contrast, our approach is designed to specifically enhance the learning of **3D geometric features without relying on external foundation models**. As shown in Supplementary Table 1, replacing the 3D geometric learning structure in I2P-MAE and ReCon with our method results in substantial performance improvements. This clearly demonstrates that our approach enhances 3D geometric representation learning in a way that effectively **complements** the semantic and textual features derived from 2D images and language foundation models.




**Motivation and Novelty of the Proposed Method**

As mentioned in the previous part, this work focuses on **3D geometric representation learning.** The motivation behind our work is twofold:

1. To highlight the limitations of using 2D images as inputs for 3D geometric learning through MAE without leveraging foundation models.
2. To emphasize the importance of incorporating multi-view information for more effective 3D geometric representation learning.

A key insight from our study is the limited effectiveness of using 2D images as input for 3D geometric learning via MAE, and the necessity of fully utilizing the multi-view attributes of 3D point clouds within the MAE framework. Our ablation studies, presented in Table 6 of the main paper and Table 3 of the supplementary material, substantiate this claim. These findings highlight an opportunity to develop more effective strategies for 3D representation learning.

The novelty of our method can be summarized as follows:

1. To the best of our knowledge, this is the first work to reconstruct both 3D and 2D information solely from 3D inputs for 3D **geometric representation learning**. A key innovation of our approach is projecting 3D point clouds to multi-view 2D images at the feature level, allowing for multi-view 2D and 3D reconstruction from a single 3D modality (see Fig. 2 in the main paper). Our projection method directly maps 3D tokens to corresponding 2D tokens for each view, ensuring seamless alignment between 3D and 2D features, thereby enabling the reconstruction of original 2D depth images from point cloud data alone.


2. We introduce a multi-scale, multi-head attention (MSMH) mechanism that enhances the interplay between local and global perspectives within each decoder transformer block, improving the quality of learned representations.

3. We propose a two-stage self-training strategy with latent space prediction, integrating latent space prediction into the MAE task to ensure that student models learn well-aligned representations during training.

It is important to note that during the fine-tuning and inference stages, we remove these additional components like projection layers and MSMH decoder, retaining the **same architecture as the baseline** to ensure a fair comparison.

---

### Meta-Review · Area_Chair_32hh · 2024-12-20

**Metareview:**

The paper proposes a self-supervised learning method for 3D point clouds. The proposed method can be combined with current Masked Autoencoder (MAE) methods to improve their performance and can utilize multi-modal information for pre-training without relying on 2D encoders. Using a two-stage self-training strategy, the proposed method achieves strong results.
However, this work has several significant drawbacks:
1. Lack of Comprehensive Comparison: the comparison with the latest methods is not comprehensive. Specifically, the paper does not fully evaluate against recent methods in 3D point cloud pre-training, such as I2P-MAE and ReCon.
2. Unclear Motivation: the rationale for not utilizing 2D foundation models is not well-justified. As demonstrated by methods such as I2P-MAE and ReCon, incorporating 2D foundation models can significantly enhance the performance of 3D point cloud pre-training.
3. Incremental Improvements: the proposed Multi-Scale Multi-Head (MSMH) module is an incremental improvement over multi-head attention mechanisms, rather than a substantial innovation. Additionally, the two-stage self-training strategy bears similarities to existing methods like iBot [1] and dBot [2].
4. Lack of Self-Containment: key explanations and experiments that support the motivations and claims are missing from the main paper, making it less self-contained.

Although the paper clearly has merit, the decision is not to recommend acceptance at this time. We encourage the authors to carefully consider the reviewers' comments when revising the paper for submission elsewhere, particularly in providing a more comprehensive comparison with the latest methods, clarifying the motivation and unique contributions, and ensuring the main paper is self-contained with all key explanations and supporting experiments.

[1] Zhou J, et al. ibot: Image bert pre-training with online tokenizer. ICLR 2022.
[2] Zhou J, et al. Exploring Target Representations for Masked Autoencoders. ICLR 2024.

**Additional Comments On Reviewer Discussion:**

This paper was reviewed by five experts in the field and finally received diverse scores: 3, 6, 8, 8, and 5.
The major concerns of the reviewers are:
1.	the technical novelty,
2.	the motivation of not utilizing 2D foundation models,
3.	the experiments conducted in the main paper are not comprehensive enough.

The authors didn’t successfully address these two concerns during the discussion period. I fully agree with these concerns and, therefore, make the decision to reject the paper.

---

### Decision · Program_Chairs · 2025-01-22

Reject